# Automated design of multi-target ligands by generative deep learning

Laura Isigkeit [1], Tim Hörmann [2], Espen Schallmayer[1], Katharina Scholz [2], Felix F. Lillich [1,3], Johanna H. M. Ehrler [1], Benedikt Hufnagel[1], Jasmin Büchner[1], Julian A. Marschner[2], Jörg Pabel [2], Ewgenij Proschak[1,3] & Daniel Merk [1,2] ✉

Generative deep learning models enable data-driven de novo design of molecules with tailored features. Chemical language models (CLM) trained on string representations of molecules such as SMILES have been successfully employed to design new chemical entities with experimentally confirmed activity on intended targets. Here, we probe the application of CLM to generate multi-target ligands for designed polypharmacology. We capitalize on the ability of CLM to learn from small fine-tuning sets of molecules and successfully bias the model towards designing drug-like molecules with similarity to known ligands of target pairs of interest. Designs obtained from CLM after pooled fine-tuning are predicted active on both proteins of interest and comprise pharmacophore elements of ligands for both targets in one molecule. Synthesis and testing of twelve computationally favored CLM designs for six target pairs reveals modulation of at least one intended protein by all selected designs with up to double-digit nanomolar potency and confirms seven compounds as designed dual ligands. These results corroborate CLM for multi-target de novo design as source of innovation in drug discovery.

Many prevalent and severe diseases such as the metabolic syndrome (MetS) and chronic inflammatory disorders are multifactorial and involve dysregulation of several signaling systems or metabolic pathways (Fig. 1a)[1–5]. While typical treatment regimens of multifactorial diseases and multimorbidity heavily rely on drug combinations[2,6,7] (also referred to as polypharmacy), designed polypharmacology, i.e., the development of molecules intentionally and simultaneously modulating more than one target involved in a disease, can achieve synergies, improve therapeutic outcomes, and overcome multi-drug treatment[8,9]. However, designing molecules that intentionally exhibit potent activity on several proteins of interest is challenging since compliance with pharmacophores for two or more target binding sites has to be achieved simultaneously[8,9]. Machine learning is increasingly used to accelerate innovation in drug discovery[10] and might also aid the development of designed multi-target drugs.

Chemical language models (CLM, Fig. 1b)[11–13] are deep learning models trained to capture string representations of molecules (i.e., the Simplified Molecular Input Line Entry System[14], SMILES) and can be used to automatically design molecules with desired properties[15–18]. CLM have been successfully employed for de novo design of new chemical entities with intended bioactivity[19–22] and enable navigation in the chemical space to obtain tailored molecular designs[11,23]. The data-driven de novo design approach using CLM is "rule-free" as CLM are capable of extracting relevant features, e.g., for bioactivity from the training molecules[23]. Therefore, CLM may also serve the task of multi-target design by accessing regions of the chemical space common to ligands of two proteins.

The development of a CLM typically proceeds in two steps[20,24]. Initial pretraining with a large collection of molecules allows the model to capture the syntax of SMILES. The pretrained general CLM is subsequently fine-tuned by transfer learning[25] with a small set of

[1]Goethe University Frankfurt, Institute of Pharmaceutical Chemistry, 60438 Frankfurt, Germany. [2]Ludwig-Maximilians-Universität München, Department of Pharmacy, 81377 Munich, Germany. [3]Fraunhofer Institute for Translational Medicine and Pharmacology ITMP, 60596 Frankfurt, Germany. ✉e-mail: daniel.merk@cup.lmu.de

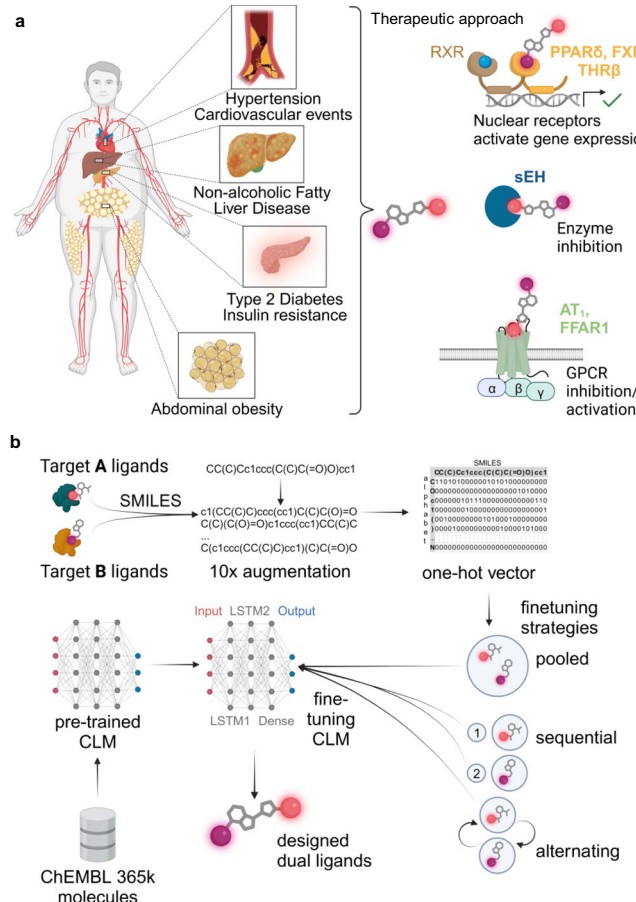

**Fig. 1 | Metabolic syndrome and application of a chemical language model for multi-target de novo design. a** The metabolic syndrome (MetS) is a multifactorial disease. Therapeutic effects against various aspects of the MetS can be achieved by nuclear receptor activation, enzyme inhibition and G-protein-coupled receptor (GPCR) modulation strategies. Designed polypharmacology addressing multiple mechanisms may provide synergies. **b** Chemical language models (CLM) are trained on molecules in string representation such as SMILES and can design new molecules in a data-driven fashion. Pretraining with a large set of molecules (e.g., from ChEMBL) allows the model to capture the syntax of SMILES. Fine-tuning with small sets of molecules can then be performed by transfer learning to bias the CLM toward designing molecules of interest. Here, we applied CLM to multi-target design by using sets of known ligands of six target pairs of interest for fine-tuning. **a, b** created with BioRender.com, released under a Creative Commons Attribution-NonCommercial-NoDerivs 4.0 International license.

molecules comprising the intended features which introduces a bias towards a region of interest in the chemical space[11,23]. The fine-tuned model can then be employed to design molecules with desired properties de novo. The critical step of fine-tuning has been successfully implemented with one to tens of known active molecules[22]. This ability of CLM to capture relevant features for bioactivity purely from molecular templates and to work in low-data regimes suggests great potential to enable data-driven multi-target design.

In this study, we systematically evaluated and optimized the application of CLM to design dual ligands for protein pairs of interest. We focused on combinations of six targets with a validated role in MetS associated disorders: (1) angiotensin II receptor type 1 ($AT_1$), which exhibits a key signaling role in the cardiovascular system[26], (2) farnesoid X receptor (FXR), a crucial hepatoprotective transcription factor and metabolic regulator[27], (3) free fatty acid receptor 1 (FFAR1, GPR40), which is involved in the control of insulin and gut hormone secretion[28], (4) peroxisome proliferator-activated receptor δ (PPARδ), a major regulator of lipid and glucose metabolism in multiple tissues[29],

(5) thyroid hormone receptor β (THRβ), which is under evaluation as a target in non-alcoholic fatty liver disease[30], and (6) soluble epoxide hydrolase (sEH), which is implicated in cardiovascular and inflammatory diseases[31,32]. These targets cover a broad range of protein families, i.e., G-protein coupled receptors ($AT_1$, FFAR1), nuclear receptors (FXR, PPARδ, THRβ) and enzymes (sEH), and ligands of these proteins have therapeutic relevance or were clinically evaluated for diseases related to the MetS. Of note, some approved drugs selectively targeting one of these proteins are regularly used together in combination therapy[33] underlining the potential of designed dual ligands in MetS.

We trained CLM to generate dual ligands of six target pairs and synthesized top-ranking candidates for prospective validation. All twelve CLM designed compounds exhibited biological activity on at least one of the intended targets and seven dual modulators were successfully obtained for three target pairs. These results underscore the value of CLM to access desired regions of the chemical space and corroborate their application for automated de novo design of multi-target ligands.

## Results

### CLM capture molecular features of pooled ligand classes

To estimate the potential of pairs of the targets of interest to bind a common dual ligand, we evaluated the chemical space covered by their known modulators. Of note, while the numbers of known ligands for the targets of interest were high (Fig. 2a), dual ligands have only been annotated for FXR/sEH (34; potency cutoff <100 μM) and FXR/THRβ (1), and the pairwise similarity between known ligand sets for the targets of interest was low (Fig. 2b). Despite high chemical diversity within all known ligand sets for the targets of interest ($EC_{50}/IC_{50} \leq 1$ μM; mean ± SD pairwise Tanimoto similarity computed on Morgan fingerprints[34]: 0.17 ± 0.02 (sEH) to 0.19 ± 0.03 ($AT_1$)), the ligands of all six targets populated defined and distinct regions of the chemical space (Fig. 2c) as illustrated by t-distributed stochastic neighbor embedding (t-SNE) with Morgan fingerprints or the fuzzier pharmacophore descriptors Chemically Advanced Template Search (CATS)[35]. Among the target pairs, an overlap was evident for FXR agonists and sEH inhibitors as well as FXR and THRβ agonists. Additionally, PPARδ agonists and sEH inhibitors populated proximal regions of the chemical space while ligands of FFAR1, $AT_1$, and sEH were more distant and hence less similar. This was also evident from scaffold analysis indicating that known ligands of FXR and sEH as well as PPARδ and sEH (potency cutoff <100 μM) shared a higher number and more complex (higher no. of atoms) scaffolds than for the other target pairs (Fig. 2d). These results pointed to a substantially higher potential to obtain dual ligands for FXR/sEH, FXR/THRβ, and PPARδ/sEH than for other pairs prompting us to focus on these combinations first.

We intended to achieve multi-target design with CLM using tailored template molecule sets covering ligands for a target pair for fine-tuning. Thus, we retrieved known binders of the proteins of interest from BindingDB[36] and clustered the ligands of each target based on their fingerprint similarity. Only the most potent compound from each cluster was kept to cover the entire known ligand space for each target and promote chemical diversity of the training molecules. After manual validation of the intended biological activity (e.g., agonism vs. antagonism) and binding mode (e.g., orthosteric vs. allosteric), 5–9 compounds per target were obtained as fine-tuning sets (Supplementary Table 1). The selected fine-tuning molecules were chemically diverse (Fig. 2e) and represented potent modulators of the targets of interest (Fig. 2f). All templates had no annotated dual activity on the studied targets.

For multi-target design model development, we built on a previously published[20] CLM which was pretrained on 365k molecules from ChEMBL[37] to capture SMILES syntax and general molecular properties. To achieve multi-target design, this general model was fine-tuned with the template molecule sets for a target pair. Three fine-tuning

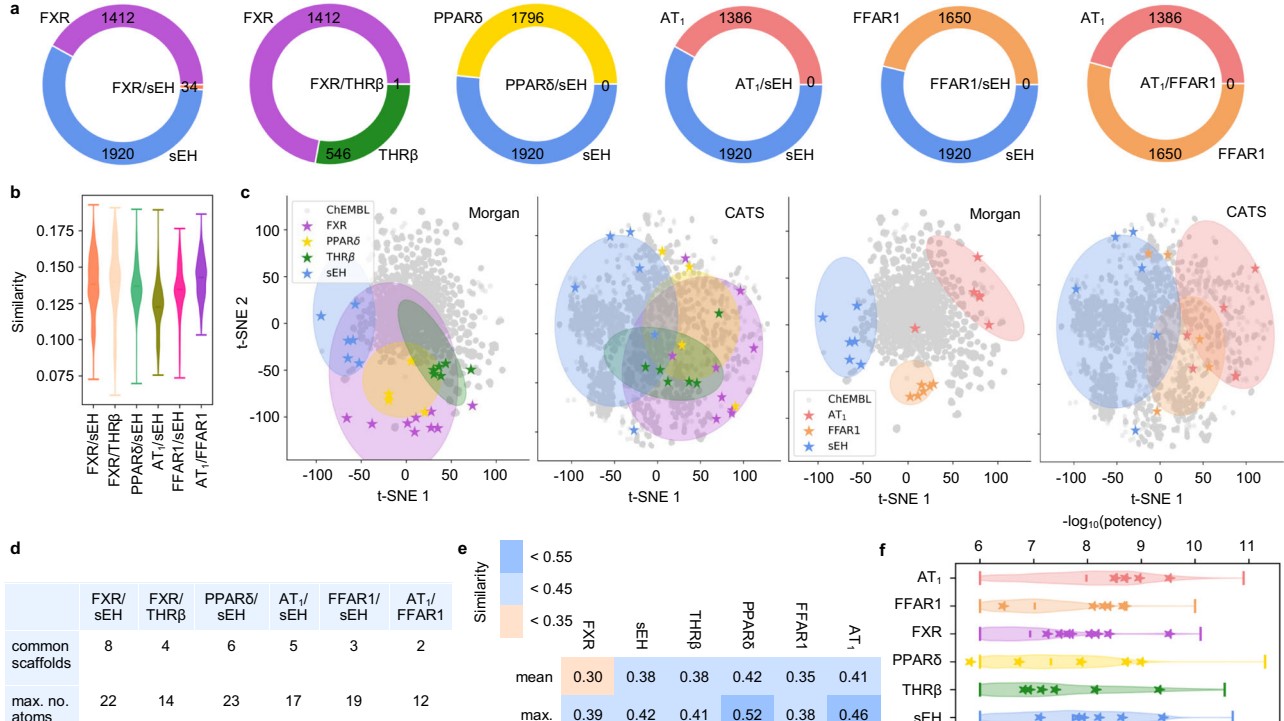

**Fig. 2 | Analysis of known ligands for targets of interest. a** Numbers of known ligands for the targets of interest (potency cutoff <100 μM) annotated in ChEMBL. Each plot represents a target pair of interest. **b** Violin plots of pairwise Tanimoto similarity distribution between ligands for the target pairs of interest. Numbers of ligands for FXR: 1412, sEH: 1920, PPARδ: 1796, THRβ: 546, AT$_1$: 1386, FFAR1: 1650. **c** Relative distribution of known ligands for the six targets of interest (FXR, sEH, THRβ, PPARδ, AT$_1$, FFAR1) illustrated as t-distributed stochastic neighbor embedding (t-SNE) (computed on Morgan fingerprints or Chemically Advanced Template Search (CATS) descriptors) with 10,000 random ChEMBL molecules as background. Known ligands of the target pairs FXR/sEH, FXR/THRβ and PPARδ/sEH populate proximal regions of the chemical space whereas AT$_1$/sEH, FFAR$_1$/sEH and AT$_1$/FFAR1 modulators are more distant (modulators with potency ≤1 μM, shaded areas). The selected fine-tuning molecules are highlighted as stars. **d** Numbers of common scaffolds in the known ligands (potency cutoff <100 μM) of the target pairs annotated in ChEMBL and their max. number of atoms. **e** Mean and max. Tanimoto similarity of molecules selected for the fine-tuning sets for the targets of interest. **f** Violin plots of potency distribution of known ligands for the targets of interest. Compounds selected for fine-tuning are highlighted as stars. The lines represent the 1st and 3rd quartiles and the mean of the distribution. Numbers of ligands with potency ≤1 μM for FXR: 693, sEH: 1601, PPARδ: 1019, THRβ: 370, AT$_1$: 1232, FFAR1: 1048. Source data are provided as a Source Data file.

strategies, (1) sequential, (2) alternating and (3) pooled fine-tuning, were evaluated using beam search[20] (width 50) to identify the best performing approach (Fig. 3a and Supplementary Fig. 1a). Beam search sampling is a heuristic method to reveal molecules with high probability to be sampled from a CLM and proved valuable to monitor fine-tuning effects[20,22]. Evaluation of beam search designs sampled during the transfer learning procedure for similarity to the fine-tuning molecules demonstrated substantial differences in the fine-tuning strategies.

Sequential fine-tuning successfully biased the model towards the first target of each pair within few epochs. However, similarity to known ligands of the first target was also rapidly lost when template molecules for the second target were introduced. Alternating fine-tuning had a similar effect with higher frequency and flipped the bias to either of the targets from epoch to epoch. The intended balanced similarity to both targets of a pair was only achieved by pooling both fine-tuning sets.

De novo designs obtained from the CLM after pooled fine-tuning exhibited favorable validity, novelty and uniqueness (Supplementary Tables 2 and 3) and had quantitative estimation of drug-likeness (QED) scores[38] (Fig. 3b and Supplementary Fig. 1b) in the range of the template molecules suggesting that the multi-target design approach by CLM generated novel drug-like molecules. With an increasing number of fine-tuning epochs, the designs approached the pooled template sets in terms of basic features such as molecular weight (MW), clogP and topological polar surface area (TPSA; Fig. 3b and Supplementary

Fig. 1b). A bias towards the chemical space around and between the template molecules after fine-tuning was also evident by visualization with t-SNE (Fig. 3c) which revealed that beam search designs from the selected epochs populated a region between known ligands for the targets of interest. Moreover, external target prediction using the Similarity Ensemble Approach (SEA)[39] demonstrated substantially enhanced probability for interaction with the targets of interest after fine-tuning compared to baseline designs from epoch 0 (Fig. 3d). Synthetic accessibility[40] of the designs was not compromised by pooled fine-tuning and remained in the favorable range of the template molecules (Fig. 3e).

## CLM designed dual ligands modulate both intended targets

With these encouraging results, we set out to test the potential of CLM based multi-target design prospectively. From the dual-target CLMs obtained by pooled fine-tuning on ligand sets for FXR/sEH, FXR/THRβ and PPARδ/sEH, we identified epochs complying best with the design criteria, i.e., balanced similarity to the fine-tuning sets for both targets (epochs 51–55 for PPARδ/sEH; cf. Fig. 3c), and sampled 5000 designs by temperature sampling[23]. From each virtual collection, we selected twelve designs based on sampling frequency[22] as model-intrinsic approach for automated design prioritization without external scoring, and twelve designs based on fingerprint similarity to the fine-tuning molecules for further evaluation (Supplementary Table 4). Three compounds per target pair were prioritized from these collections by docking for prospective validation by synthesis and in vitro

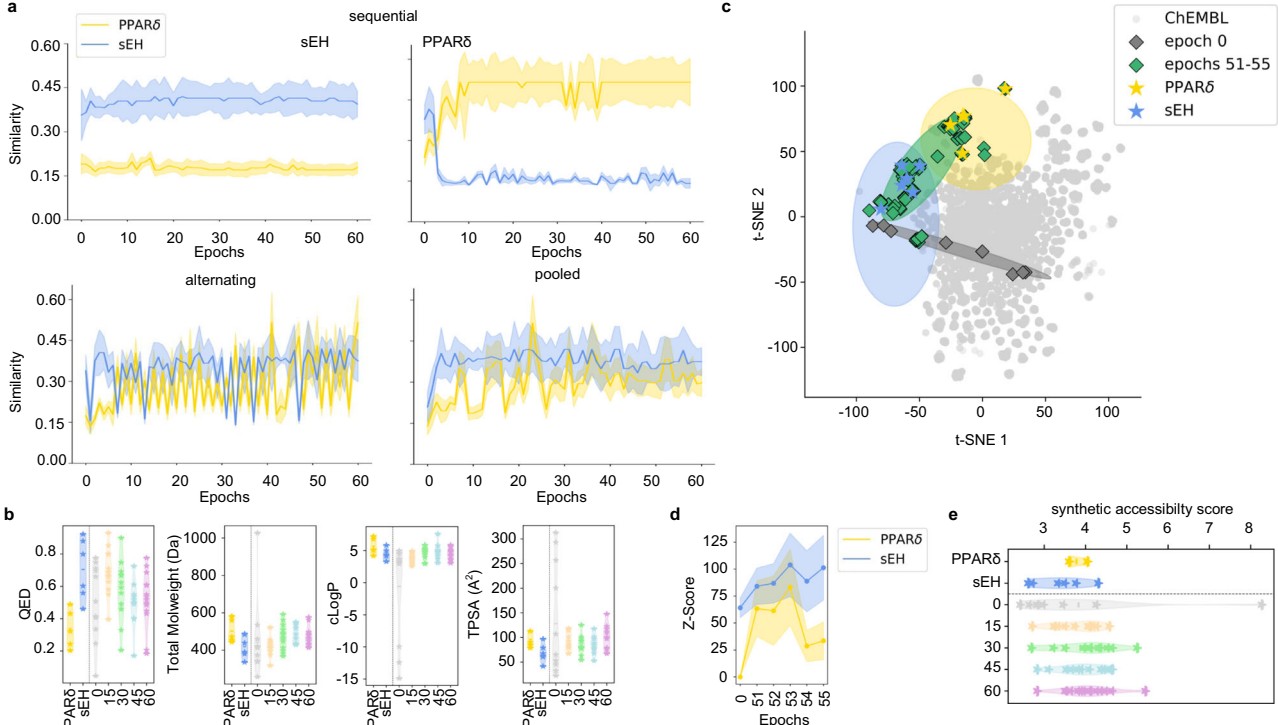

**Fig. 3 | Fine-tuning of chemical language models (CLM) for multi-target design.** The target pair PPARδ/sEH is shown as example in (**a**–**e**). **a** Effects of different CLM fine-tuning strategies on the similarity of beam search designs (width 50) to the fine-tuning molecules. Fine-tuning with pooled template sets was superior to sequential and alternating fine-tuning strategies in terms of design similarity to both fine-tuning collections. Graphs show the max. Tanimoto similarity ± SD (standard deviation of max. Tanimoto similarity) computed on Morgan fingerprints of the beam search designs per epoch to the fine-tuning molecules. For each epoch beam search designs (width 50) were generated and only valid SMILES were analyzed. **b** Quantitative estimation of drug-likeness (QED) scores[38] and basic molecular features of beam search designs over the fine-tuning procedure (epochs 0, 15, 30, 45, 60) illustrated as violin plots. Stars represent the fine-tuning molecules and beam search designs, respectively. Numbers of analyzed ligands/designs PPARδ: 5, sEH: 6, epochs 0: 15, epoch 15: 23, epoch 30: 17, epoch 45: 10, epoch 60: 13 (only valid SMILES from beam search were

valid SMILES from beam search were analyzed). **c** Effects of pooled fine-tuning on design features (visualized as t-distributed stochastic neighbor embedding (t-SNE)). Top beam search designs from the epochs of highest similarity to both fine-tuning sets (51–55) in comparison to beam search designs from epoch 0 are shown. Numbers of analyzed ligands/designs PPARδ: 5, sEH: 6, epoch 0: 15, epochs 51–55: 249. **d** Target prediction (Z-scores) of designs from epochs 51–55 using the Similarity Ensemble Approach (SEA)[39] for the targets of interest. Z-Scores are the mean ± standard deviation (SD). **e** Synthetic accessibility scores[40] for the beam search designs obtained from different epochs during pooled fine-tuning visualized as violin plots. Fine-tuning molecules and beam search designs are highlighted as stars, respectively, and the middle line represents the mean of the distribution. Numbers of analyzed ligands/designs PPARδ: 5, sEH: 6, epochs 0: 15, epoch 15: 23, epoch 30: 17, epoch 45: 10, epoch 60: 13 (only valid SMILES from beam search were analyzed). Source data are provided as a Source Data file.

characterization. Two compounds (**1**, **2**) for the FXR/sEH target pair resembled previously developed dual FXR/sEH modulators[41] thus strongly pointing to dual activity. Of note, these previously known dual FXR/sEH ligands were not used to develop the model. As an additional proof of concept, we applied the design approach to the FXR/PPARδ target pair (Supplementary Table 5) for which dual agonists have been previously obtained by rational design[42]. A CLM fine-tuned with the pooled selective FXR and PPARδ ligand template sets (not containing dual ligands) successfully generated known dual FXR/PPARδ agonists and several structurally similar molecules supporting the applied design strategy.

The CLM designed dual ligand candidates **3**–**9** were successfully synthesized over 2–6 steps with 1–68% overall yield (Fig. 4) demonstrating that the dual target CLM generated synthesizable molecules. Compound **7** was originally designed with *tert*-butyloxy motif which was replaced by *tert*-butyl for improved chemical stability. The dual ligand candidates were characterized in functional assays for the targets of interest using cellular Gal4-hybrid reporter gene assays[43] to determine FXR, THRβ and PPARδ modulation and an enzyme activity assay[44] with the fluorogenic substrate PHOME to test sEH inhibition (Table 1). In vitro characterization of the CLM designed dual FXR/sEH modulators **1**–**3** confirmed engagement of both targets by all three compounds with sub-micromolar to low micromolar potencies. The dual FXR/THRβ

agonist designs obtained from the model exhibited substantial FXR agonism combined with weak (**6**) or no detectable (**4**, **5**) THRβ agonism at non-toxic concentrations. All designed PPARδ/sEH modulators (**7**–**9**) were confirmed as dual ligands of the intended targets. **7** emerged as remarkably potent sEH inhibitor endowed with micromolar PPARδ agonist activity. **8** and **9** were less potent on both targets.

These results from in vitro testing corroborated multi-target design by CLM. Seven out of nine dual ligands designed by the models modulated both intended targets. With potencies varying between 17 nM and >100 μM, and mostly unbalanced activity on the respective target pair, the CLM designs require structural optimization, but their scaffolds are highly valuable leads for designed polypharmacology.

## CLM designs fuse molecular and pharmacophore features of ligands for two targets

The predicted binding modes of the dual modulators **1**, **6** and **7** to the targets of interest (Fig. 5a–c) revealed key interactions that were also formed by the co-crystallized ligands. The designs thus complied with the pharmacophore requirements of the respective binding sites indicating that the CLM could capture relevant pharmacophore elements for the respective two binding sites and fuse them in one molecule. This was also evident from structural comparison of the

CLM designed dual ligands with the most similar ligands annotated in ChEMBL for the respective targets (Fig. 5d). The dual FXR/THRβ modulator **6** combined the well-known 5-alkyl-3-phenylisoxazole motif of FXR agonists with a (4-benzylphenoxy)acetic acid residue inherited from THRβ agonists. The latter group at the same time resembled aromatic linkers found in FXR agonists bearing the essential carboxylic acid motif. Similarly, the dual PPARδ/sEH ligand **9** comprised a hydrophobic urea scaffold widely found in sEH inhibitors which was merged with a 3-(4-alkyloxyphenyl)propanoic acid pharmacophore of PPARδ agonists. Dual ligands designed by the CLM thus comprised merged pharmacophores[8,9], were drug-like with similar QED scores as the template molecules and exhibited preferable synthetic accessibility (Fig. 5e and Supplementary Fig. 2). Due to the structural fusion of ligand features for the proteins of interest, the designed dual ligands exhibited similarity to known modulators of both their targets (Supplementary Table 6) and resembled the respective fine-tuning sets in basic properties like MW, clogP and TPSA (Fig. 5f and Supplementary Fig. 2).

Comparison of the designed dual ligands with the fine-tuning molecules (Fig. 6a and Supplementary Fig. 3) indicated that the CLM fused pharmacophore elements with varying degrees of structural diversity. The designs exhibited low to intermediate Tanimoto similarity to their respective most similar fine-tuning templates and Rapid Similarity Calculation of Maximum Common Edge Subgraph (RascalMCES)[45], which identify maximum common sub-skeletons of pairs of molecules, revealed substructure contributions from both selective templates to the dual modulators. To verify that the polypharmacology of the dual ligands was indeed due to the pharmacophore fusion and structural modification designed by the CLM, we tested the commercially available fine-tuning compounds for PPARδ/sEH as representative target pair in the same assays as the designed dual ligands (Table 2). The evaluated fine-tuning molecules were selective for their annotated bioactivity over the second target of the respective designs providing further support for the ability of CLM to fuse selective pharmacophores to dual ligands. Although a potential bias from other molecules used to (pre-)train the CLMs, which act as dual ligands but have not been tested on the targets of interest, cannot be fully excluded, these results indicate that structural fusion and moderate structural modification by the CLM compared to the selective templates was driving the designed polypharmacology.

To validate our design approach further, we evaluated whether the CLM learned target-specific SAR features from the pooled dual fine-tuning by comparing with designs from a native CLM without fine-tuning and with designs obtained after fine-tuning with the pooled templates for all four targets (Fig. 6b–d). In contrast to pooled dual fine-tuning, the Tanimoto similarity to the template sets did not consistently increase when the CLM was fine-tuned with the pooled templates for FXR, THRβ, PPARδ and sEH (Fig. 6b) suggesting potentially that simultaneous training with molecular information for four targets was too diverse to extract the relevant structural features or too complex to be structurally fused. Similarly, target prediction for the designs obtained after pooled fine-tuning for four targets revealed increasing scores but no consistent balanced improvement and strong variance over the fine-tuning procedure (Fig. 6c).

We then used pharmacophore models for the targets of interest (Fig. 6d and Supplementary Fig. 4) to estimate the rates of designed dual ligands obtained from a native baseline CLM, after pooled fine-tuning for target pairs (dual), and after fine-tuning with the pooled templates for all four targets. The fraction of designs from the baseline CLM matching both pharmacophore models of a target pair of interest was small (Fig. 6d) aligning with very low scores in the target prediction for the native model (Fig. 6c, epoch 0—baseline model). Pooled fine-tuning with the templates for all four targets slightly enhanced the fraction of designs matching pharmacophore models for the target pairs but was inferior to the focused dual models. These results further

corroborated the pooled fine-tuning for target pairs to achieve dual ligand design by CLM.

## Dual modulator design by CLM fails for target pairs with highly diverse ligands

Intrigued by the favorable results for CLM-driven dual ligand design we next addressed the target pairs whose known ligands populated more distant regions in the chemical space (Fig. 2c). Using the same procedures that successfully generated dual FXR/sEH, FXR/THRβ, and PPARδ/sEH modulators, we designed dual ligand candidates for $AT_1$/sEH, FFAR1/sEH, and $AT_1$/FFAR1 and selected one design per target pair for synthesis based on sampling frequency and subsequent docking. Again, the computationally favored designs **10**−**12** were synthesizable with good overall yields (14–40% over 4 steps; Fig. 7a). For improved accessibility, the dual FFAR1/sEH modulator design **11** was slightly modified by replacing a cyclopropyl substituent suggested by the CLM with an ethyl group. In vitro testing in cellular assays for $AT_1$ and FFAR1, and an enzyme activity assay for sEH revealed biological activity for all three designs (Table 3) but no design exhibited the intended dual modulation. The dual $AT_1$/sEH (**10**) and FFAR1/sEH (**11**) modulator candidates potently inhibited sEH but were inactive on the intended GPCRs. The dual $AT_1$/FFAR1 design **12** activated FFAR1 but showed no effect on $AT_1$ activity. Despite only testing one design per target pair, these results may suggest that the CLM was able to capture features relevant for modulation of the targets of interest but failed to access a chemical space region of dual ligands. However, based on the different distribution of known modulators for these targets in the chemical space (cf. Fig. 2c) it can be speculated whether such common region exists. Of note, ChEMBL contains no molecule with an annotated dual activity at <100 μM on $AT_1$/sEH, FFAR1/sEH or $AT_1$/FFAR1 (cf. Fig. 2a). The failure to access a chemical space of dual ligands for the targets of interest could also be observed by target prediction (SEA, Fig. 7b) which suggested improvement for only one target (FFAR1/sEH) or varying performance ($AT_1$/FFAR1). Only the dual $AT_1$/sEH focused designs tended to have a higher probability for activity on both targets at later fine-tuning epochs.

## Discussion

Generative deep learning is on the rise as a tool to design innovative new chemical entities (NCEs) with tailored properties such as bioactivity on an intended target. Such de novo design by deep learning is data-driven and overcomes the need for predefined rules, thus offering potential access to a wider chemical space[11,23]. Among various graph-based and sequence-based approaches[19–22,46–49], CLM have been particularly successful in designing NCEs with experimentally confirmed activity which was achieved by fine-tuning with known ligands for the targets of interest as templates. Based on this valuable performance of CLM in transfer learning even in low-data scenarios[22,23], we here sought to extend the scope of CLM application to multi-target design, i.e., de novo design of NCEs with biological activity on two intended targets. Designed multiple ligands of proteins that are co-involved in one pathology can exhibit enhanced efficacy by exploiting synergistic effects and may be valuable for drug development. For example, in multifactorial pathologies like the MetS or chronic inflammatory diseases, designed polypharmacology could be an avenue to improve therapeutic outcomes and overcome polypharmacy. However, the development of multi-target ligands is challenging and typically requires sophisticated design and extensive optimization. CLM could accelerate solving this task as an innovative data-driven approach to designed polypharmacology.

CLM learn from data and can capture higher-order chemical features from the molecular string representation of SMILES to achieve rule-free de novo design. Fine-tuning with a tailored set of molecules comprising the desired features is the critical step in CLM development and allows biasing the model towards a chemical space region of

**Fig. 4 | Synthesis of the dual ligand candidates 1–9 designed by the chemical language model (CLM).** The CLM designed synthesizable and dual active ligands (Table 1) for FXR/sEH, FXR/THRβ and PPARδ/sEH.

interest. We reasoned that data-driven de novo design by CLM could offer access to multi-target ligands by employing sets of known modulators for target pairs of interest for fine-tuning. Pooled sets of template ligands indeed successfully biased the model to design molecules exhibiting similarity to both template sets. Sequential fine-tuning, in contrast, produced a bias towards one target per pair and the alternating approach seemed to rather disturb the model. Designs obtained from CLM after pooled fine-tuning resembled the combined fine-tuning sets in terms of basic molecular features and had higher predicted probabilities for interaction with the targets of interest than designs from the baseline model.

We selected multi-target de novo designs, that were computationally favored based on similarity and on sampling frequency as a model intrinsic measure, for experimental validation. Synthesis and in vitro evaluation confirmed all twelve tested designs as active on at least one of the targets of interest and seven out of twelve modulated both intended targets with varying potencies. This high success rate in obtaining dual ligands may be due to a focus on target pairs with well compatible binding sites and is likely not representative of a broader application of CLM for multi-target design. Still, this outcome demonstrates potential of CLM in designing dual modulators as innovative bioactive NCEs.

Experimentally confirmed dual ligands were successfully obtained for FXR/sEH, FXR/THRβ, and PPARδ/sEH while the computationally most favored designs obtained from the CLM for AT$_1$/sEH, FFAR1/sEH, and AT$_1$/FFAR1 did not act as dual modulators but were selective for one of the targets of interest. Known ligands of target pairs for which dual ligands were successfully obtained populate proximal regions in the chemical space indicating higher chemical similarity compared to ligands of AT$_1$/FFAR1 and sEH. Despite only testing one design per target pair for the less similar combinations, these observations indicate that de novo design of multi-target ligands by CLM is feasible in scenarios where the template molecules representing the targets of interest are close in the chemical space and more challenging for target combinations whose ligands are chemically more diverse—a scenario that also hinders multiple ligand development by traditional, systematic means[8]. While target pairs with too diverse ligand binding sites will likely not be accessible for designed polypharmacology due to their inability to bind a common ligand, dual modulation of target combinations with intermediate ligand similarity might be achievable with further improved CLM, combined approaches like interactome learning[50], or newer models like GPT.

Comparison with the baseline model and a CLM fine-tuned with all template molecules confirmed that the focused dual ligand CLMs

**Table. 1 | Activity of dual ligand designs 1–9 on the targets of interest**

| ID | Structure | In vitro biol. activity |
|---|---|---|
| 1 |  | FXR: $EC_{50}$ = 0.37 µM<br>sEH: $IC_{50}$ = 4.1 µM |
| 2 |  | FXR: $EC_{50}$ = 0.44 µM<br>sEH: $IC_{50}$ = 8.1 µM |
| 3 |  | FXR: 3.4-fold act. at 50 µM<br>sEH: $IC_{50}$ = 21 µM |
| 4 |  | FXR: $EC_{50}$ = 12 µM<br>THRβ: inactive |
| 5 |  | FXR: $EC_{50}$ = 11 µM<br>THRβ: inactive |
| 6 |  | FXR: $EC_{50}$ = 1.0 µM<br>THRβ: 5-fold act. at 15 µM |
| 7 |  | PPARδ: $EC_{50}$ = 10 µM<br>sEH: $IC_{50}$ = 0.017 µM |
| 8 |  | PPARδ: 6.7-fold act. at 80 µM<br>sEH: $IC_{50}$ = 0.20 µM |
| 9 |  | PPARδ: 3.0-fold act. at 80 µM<br>sEH: $IC_{50}$ = 0.27 µM |

At least one dual ligand was identified for all three target pairs. Activities were determined in Gal4-hybrid reporter gene assays (FXR, THRβ, PPARδ) and an enzyme activity assay (sEH). Activity data are the mean; $n \geq 3$.

captured target-specific ligand features. Designs with confirmed dual activity obtained from these dual ligand CLM comprised structural contributions from template ligands for both targets highlighting an ability of CLM to merge pharmacophores. In contrast to linking or fusing the chemical features relevant for activity, dual ligands with merged pharmacophores offer favorable drug-like properties and are most difficult to obtain in designed polypharmacology[8,9]. These favorable results therefore corroborate CLM for multi-target de novo design.

## Methods

### Computational methods

**Software.** Python (v3.7.13 for CLM and v3.9.16 for visualizations), RDKit (v2022.09), DataWarrior (v5.5.0), tensorflow (v2.14.0), keras (v.2.14.0) scikit-learn (v1.2.2), matplotlib (v3.8.4), pandas (v.2.2.2) and numpy (v.1.26.4) in Python (v3.7.13). Molecular Operating Environment[51] (MOE, version 2022.02, Chemical Computing Group Inc. Montreal, QC, Canada) for molecular docking and pharmacophore generation and search.

**Data processing.** Molecules were encoded as canonical SMILES using RDKit (v2022.09, www.rdkit.org). Only SMILES up to 140 characters in length were retained and standardized in Python (v3.9.16, www.python.org) by removing stereochemistry, salts, and duplicates.

**Fine-tuning sets.** Molecules with $IC_{50}/EC_{50} \leq 1\,µM$ on the targets of interest were selected from BindingDB[36] (v2021). After processing (cf. Data processing), the molecules were clustered based on Morgan fingerprints[34] (length = 1024, radius = 2) using the k-means algorithm from scikit-learn (v1.2.2, www.scikit-learn.org) in Python (v3.9.16). The cluster number was optimized from 2 to the number of molecules to minimize the silhouette score. From the resulting clusters, the molecule with the lowest $EC_{50}/IC_{50}$-value was selected as representative. From this collection, the final fine-tuning molecules for each target were selected based on manual binding mode inspection and confirmation of the desired biological activity (agonism, inhibition, etc.).

**CLM implementation.** We used a recently published framework (www.github.com/ETHmodlab/ molecular_design_with_beam_search)[20] to implement the multi-target CLM in Python (v3.7.13) using tensorflow (v2.14.0) and keras API (www.tensorflow.org, v2.14.0). The model was based on a recurrent neural network with long short-term memory (LSTM) cells and consisted of four layers with a total of 5,820,515 parameters: layer 1, BatchNormalization; layer 2, LSTM with 1024 units; layer 3, LSTM with 256 units; layer 4, BatchNormalization. The CLM was trained using the Adam optimizer (learning rate = $10^{-3}$), categorical cross-entropy loss, and a dropout rate of 0.4 for both LSTM layers, with the first layer set to frozen. The CLM was trained on SMILES strings encoded as one-hot vectors and all SMILES were augmented 10-fold.

**Training strategies.** In each strategy, the CLM was trained over 60 epochs. Pooled: both fine-tuning sets were used together in a single set to fine-tune the CLM. Alternating: the two fine-tuning sets were alternatingly presented to the CLM in each epoch. Sequential: the CLM was first trained with one fine-tuning set for 60 epochs. Then, based on the maximum similarity of the beam designs to the fine-tuning set, one epoch was selected to continue training with the other fine-tuning set.

**Beam search and epoch selection.** We used beam search[20] with a beam width of 50 to monitor fine-tuning and select epochs for sampling. The maximum SMILES string length was defined as 140 tokens.

**Temperature sampling.** SMILES were sampled using the softmax function parameterized by two different sampling temperatures (0.2 and 0.7). The probability of the $i$th character to be sampled from the CLM was computed as: $q_i = \exp\left(\frac{z_i}{T}\right) / \sum_j \exp(z_j / T)$ where $z_i$ is the CLM

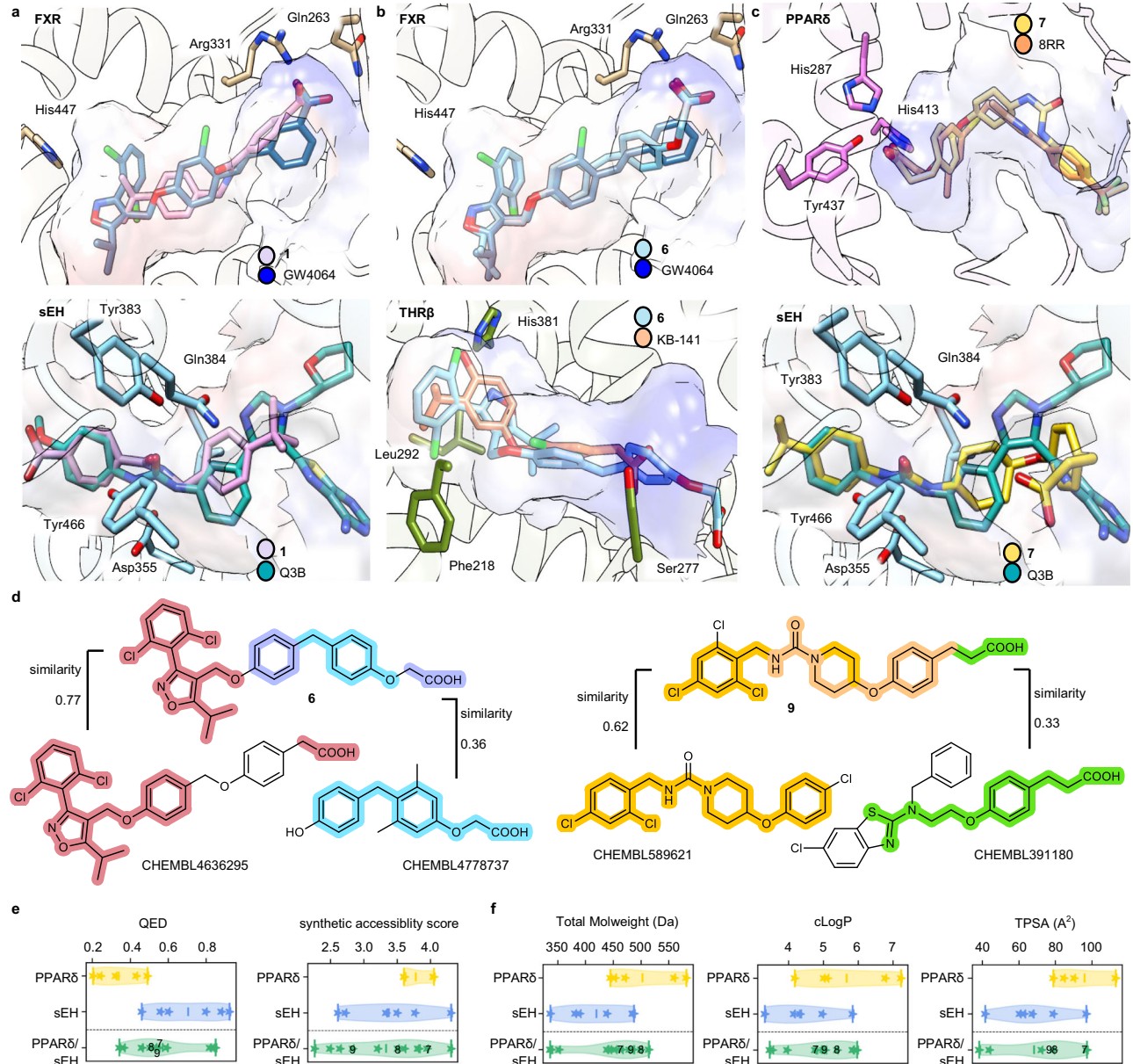

**Fig. 5 | Features of dual ligands designed by the chemical language models (CLM). a–c** Predicted binding modes of the most active dual ligands **1**, **6** and **7**. (FXR: protein data bank (pdb) ID 6a60[53], sEH: pdb ID 5ali[52], THRβ: pdb ID 1nax[54], PPARδ: pdb ID 5y7x). **d** Structural and pharmacophore comparison of the dual FXR/THRβ ligand **6** and the dual PPAR/sEH modulator **9** with the most similar ligands of the targets of interest annotated in ChEMBL (pretraining set). Similarity refers to Tanimoto similarity computed on Morgan fingerprints[34]. Common substructures in the designs and most similar ligands are highlighted (FXR−red, THRβ −blue, violet−both and yellow−sEH, green−PPARδ, orange−both). **e** Violin plots of quantitative estimation of drug-likeness (QED)[38] and synthetic accessibility scores of the CLM designs were favorable and resembled the fine-tuning molecules. Stars represent the fine-tuning molecules and designed dual ligands, respectively, the lines represent the 1st and 3rd quartiles and the mean. Numbers of the ligands/ designs for PPARδ: 5, sEH: 6, PPARδ/sEH: 12. **f** Violin plots of basic chemical features of the CLM designs resembling the fine-tuning molecules. Stars represent the fine-tuning molecules and designed dual ligands, respectively, the lines represent the 1st and 3rd quartiles and the mean. Numbers of the ligands/designs for PPARδ: 5, sEH: 6, PPARδ/sEH: 12. Source data are provided as a Source Data file.

prediction for character $i$, $T$ is the temperature, and $q_i$ is the sampling probability of character $i$.

**Top 12 selection.** For both sampling temperatures (0.2, 0.7) we selected 12 CLM designs with highest sampling frequency and 12 CLM designs with highest geometric mean similarity to their fine-tuning sets. The geometric mean between frequency and similarity led to the selection of the top 12 candidate designs.

**Molecular docking.** From the top 12 candidate designs, molecules were selected for synthesis and in vitro characterization by molecular

docking. Docking was performed in Molecular Operating Environment[51] (MOE, version 2022.02, Chemical Computing Group Inc. Montreal, QC, Canada). The ligand-bound X-ray structures of sEH (pdb ID: 5ali[52]), FXR (pdb ID: 6a60[53]), THRβ (pdb ID: 1nax[54]), PPARδ (pdb ID: 5y7x), AT₁ (pdb ID: 4zud[55]) and FFAR1 (pdb ID: 4phu[56]) served as templates. The structures were prepared using the MOE QuickPrep tool with default settings, adjusting the protonation state of the complex. Ligands were prepared using the MOE Wash tool with dominant protonation state at pH 7.0; coordinates were rebuilt 3D; existing chirality was maintained. The following settings were used for all docking calculations: Force Field = Amber10:EHT, Receptor =

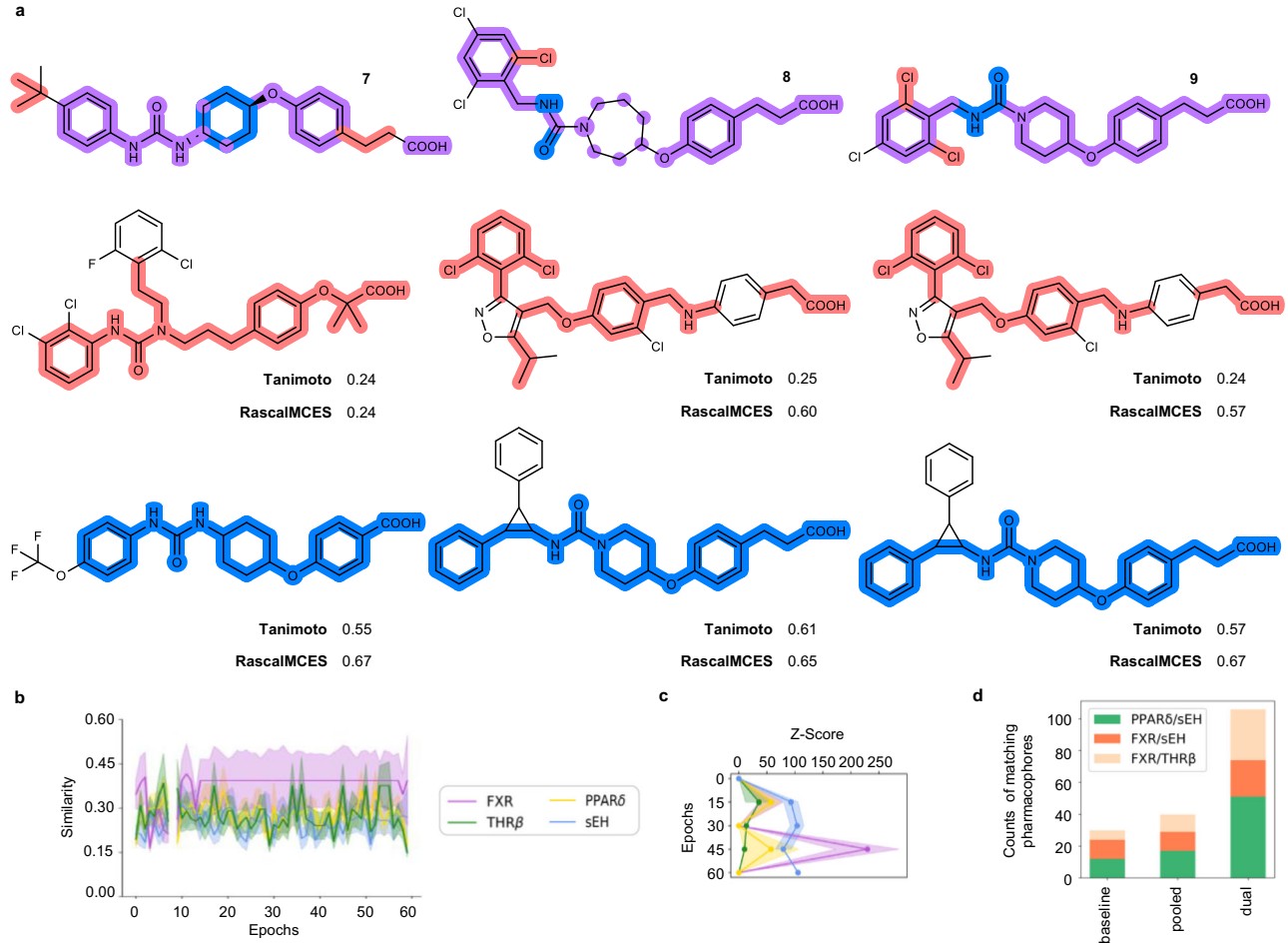

**Fig. 6 | Validation and comparison of the dual ligand design approach.**
**a** Comparison of designed dual PPARδ/sEH modulators with the most similar fine-tuning templates by Tanimoto similarity computed on Morgan fingerprints and Rapid Similarity Calculation of Maximum Common Edge Subgraph (RascalMCES)[45]. The highlighted substructures represent the extracted RascalMCES between the most similar fine-tuning templates and the designed dual modulators (red−PPARδ, blue−sEH, violet−both). **b** Effects of chemical language model (CLM) fine-tuning with the pooled templates for FXR, THRβ, PPARδ and sEH. The max. Tanimoto similarity ± SD (standard deviation of max. Tanimoto similarity) computed on Morgan fingerprints of the beam search designs (width 50) per epoch to the fine-tuning molecules for the respective target is shown. For each epoch beam search designs (width 50) were generated and only valid SMILES were analyzed. **c** Target prediction (Z-scores) of beam search designs (width 50) over the fine-tuning procedure using the Similarity Ensemble Approach (SEA)[39] for the targets of interest. Z-Scores are the mean ± SD. For each epoch beam search designs (width 50) were generated and only valid SMILES were analyzed. **d** Bar chart of fractions of designs obtained from the native baseline CLM, after fine-tuning with the pooled templates for FXR, THRβ, PPARδ and sEH, or the pooled ligands for the target pairs of interest (dual) matching pharmacophore models (Supplementary Fig. 4) of the respective target pairs. Source data are provided as a Source Data file.

Receptor and Solvent Atoms, Site = Crystallized Ligand Atoms, Placement = Template with 100 poses, Refinement = Rigid Receptor, scoring function = GBVI/WSA dG with 10 poses.

**Molecular descriptors.** Morgan fingerprints[34] (length = 1024, radius = 2), RascalMCES[45] (similarityThreshold = 0.5) and quantitative estimation of drug-likeness[38] (default settings) were computed using RDKit (v2022.09) in Python (v3.9.16). Total molweight, cLogP and TPSA were computed using DataWarrior[57] (v5.5.0). Synthetic accessibility score was computed using SwissADME[40]. CATS[35] descriptors were computed with https://github.com/iwatobipen/CATS2D.

**Target prediction.** The web interface of the Similarity ensemble approach (SEA)[39] (https://sea.bkslab.org/) was used for target prediction.

**Stochastic neighbor embedding.** The t-SNE projection was performed with scikit-learn (v1.2.2) in Python (v3.9.16) on Morgan fingerprints and on CATS[35] descriptors. The following settings were chosen: perplexity = 30, learning_rate = "auto", init = "pca".

**Pharmacophore modeling and search.** For pharmacophore modeling, the ligands of the superimposed X-ray structures of sEH (pdb IDs: 5ali[52], 3ant[58], 3wke[59]), FXR (pdb IDs: 6a60[53], 3dcu[60], 3fli[61]), THRβ (pdb IDs: 1nax[54], 1r6g[62], 6kkb[63]) and PPARδ (pdb IDs: 5y7x, 3tkm[64], 5u46[65]) were used in Molecular Operating Environment[51] (MOE, version 2022.02, Chemical Computing Group Inc. Montreal, QC, Canada). A consensus pharmacophore model was generated per target considering features present in at least 2 ligands. Binding site residues of the target were defined as excluded volumes. The sampled designs of the respective target pairs were screened with the respective pharmacophore models for the targets of interest.

**In vitro assays**
**Cell culture for stably transfected CHO-K1 cell lines.** CHO-K1 cells (DSMZ, ACC110) were cultured at 5% CO₂ and 37 °C in full growth medium (Ham's F-12 medium supplemented with 10% fetal calf serum (FCS), penicillin (100 U/mL) and streptomycin (100 μg/mL)). For harvesting cells were incubated with Trypsin/EDTA. *IP-One assay for AT₁ and FFAR1.* To investigate the ligand dependent activation of the two GPCRs, inositol mono-phosphate (IP-One) accumulation was

**Table. 2 | Activity of PPARδ/sEH fine-tuning molecules on the targets of interest**

| Fine-tuning molecule | EC$_{50}$(PPARδ) [µM] | IC$_{50}$(sEH) [µM] | Tanimoto |
|---|---|---|---|
| | 0.016 | >30 | 0.17; 0.13; 0.13 |
| | 0.022 | >30 | 0.20; 0.15; 0.16 |
| | 0.009 | >30 | 0.13; 0.11; 0.11 |
| | >50 | 0.005 | 0.55; 0.25; 0.27 |
| | >50 | 0.097 | 0.38; 0.19; 0.19 |
| | >50 | <0.001 | 0.11; 0.13; 0.13 |

Commercially available fine-tuning molecules for PPARδ/sEH were experimentally confirmed selective for their annotated bioactivity over the respective second target in the assays used to characterize the dual PPARδ/sEH ligand designs 7–9. Data are the mean; n = 3. Tanimoto refers to the fine-tuning molecules' similarity to the experimentally tested dual ligand designs 7–9.

monitored using an HTRF based displacement assay between FRET acceptor coupled IP-One and Terbium cryptate coupled anti-IP-One antibody (IP-One assay kit, Cisbio Bioassays, Codolet, France). *AT$_1$:* the IP-One assay for AT$_1$ was performed according to the protocol published by Hernandez-Olmos et al.[66] In brief, by using the sleeping beauty method a CHO-K1 cell line was generated which stably over-expresses AT$_1$ isoform 1 and GNA11 (G-protein subunit α 11) as well as cells only overexpressing the GNA11 without any GPCR, as control. Cells were seeded into white tissue culture 384-well plates (Greiner Bio-One, Frickenhausen, Germany) at 15,000 cells/well and incubated overnight at 37 °C and 5% CO$_2$. The next day the medium was removed and the cells were washed four times with stimulation buffer (146 mM NaCl, 4.2 mM KCl, 1 mM CaCl$_2$, 0.5 mM MgCl$_2$, 50 mM LiCl, 5.5 mM D-glucose, 0.1% (w/v) fatty acid-free bovine serum albumin fraction V buffered with 10 mM HEPES at pH 7.4 (NaOH)) using a Tecan Hydro-Speed plate washer (Tecan Deutschland GmbH, Crailsheim, Germany). Thereafter, the compounds and a total of 0.5% DMSO were added to the cells. The plate was sealed and incubated for 90 min at 37 °C. Afterwards the cells were lysed by addition of the detection agents prepared in lysis buffer according to the manufacturer's instructions.

To calculate the IP-One concentration produced by the cells, a standard curve using dilutions of unlabeled IP-One in buffer without cells was used. For control of assay performance, a full dose-response curve of the known agonist [Val5]-Angiotensin II (Sigma-Aldrich #A2900) was conducted. Antagonist activity of compounds was measured with parallel stimulation with 10 nM [Val5]-Angiotensin II. *FFAR1:* the cell line generation for FFAR1 was conducted according to the protocol published by Ehrler et al.[67] and the second messenger assay was performed as described for AT$_1$. A full dose-response curve with FFAR1 reference agonist GW9508 was measured as control for assay performance. All compounds were tested at the indicated concentration and CHO-K1 wt cells were used as control.

**sEH activity assay.** Soluble epoxide hydrolase (sEH) was purified as described by Hahn et al.[68] and Lukin et al.[69], respectively. Protein expression was carried out in 500 ml ZYP5052 autoinduction medium containing Kanamycin (100 mg/L) with lactose uptake-dependent induction. The fermentation procedure started at 37 °C and 200 rpm until the OD600 reached approximately 1.0. Then the temperature was reduced to 16 °C. After 24–36 h the cells (OD600 ≈ 15.0) were harvested by centrifugation at 5000 × g (4 °C, 20 min). Pellets were resuspended in a resuspension buffer containing Tris (50 mM), NaCl (0.5 M) and MgCl$_2$ (10 mM) at pH 8 and sonicated twice for 15 s on ice. The sonicated suspension was centrifuged (18,000 × g, 30 min, 4 °C) and the supernatant was ultracentrifuged (100,000 × g, 1 h 10 min, 4 °C). The ultracentrifugation supernatant was loaded on a 5 ml HisTrap® FF (GE Healthcare, München, Germany) column at a flow rate of 2 ml/min and washed with washing buffer containing Tris (50 mM), NaCl (0.5 M), MgCl$_2$ (10 mM), imidazole (20 mM) and DTT (3 mM) at pH 8, and eluted at 120 mM imidazole. The fractions containing hsEH after SDS-PAGE fraction analysis were pooled. The identity of hsEH was confirmed by Western Blotting and hsEH hydrolase activity assay. The inhibitory potency of the compounds was determined using a fluorescence-based enzyme activity assay in which PHOME (3-phenyl-cyano(6-methoxy-2-naphthalenyl)methylester-2-oxiraneacetic acid) is metabolized by sEH hydrolase towards a fluorescent naphthalene aldehyde[70]. Dilution series of the compounds were incubated for 30 min with recombinant sEH (final protein concentration 3 nM). Subsequently, the fluorogenic substrate PHOME at final concentration of 50 µM was added, and the fluorescence was measured every minute for 45 min using a Tecan plate reader (ex: 360 nm, em: 465 nm; bandwidth 35 nm). Inhibition in percent was plotted against the logarithmic test compound concentration and IC$_{50}$-values were calculated using the "log(Inhibitor) vs. Response – Variable slope (four parameters)" equation in GraphPad Prism 7.

**Hybrid Gal4 reporter gene assays.** HEK293T cells (ATCC, CRL-1573) were cultured in Dulbecco's modified eagle medium (DMEM), high glucose, supplemented with 10% FCS, sodium pyruvate (1 mM), penicillin (100 U/mL), and streptomycin (100 µg/mL) to 70–80% confluence at 37 °C and 5% CO$_2$ and seeded in clear 96-well plates with a density of 40,000 cells/well. After 24 h, the medium was changed to Opti-MEM without supplements. Transient transfection was performed using Lipofectamine LTX reagent (Invitrogen, Carlsbad, CA, USA) according to the manufacturer's protocol with pFR-Luc (100 ng/well) as reporter plasmid, pRL-SV40 (2 ng/well) for normalization of transfection efficiency and cell growth, and one pFA-CMV-hNR-LBD clone coding for the hinge region and ligand binding domain (LBD) of the NR of interest. Five hours after transfection, the cells were incubated with Opti-MEM supplemented with penicillin (100 U/mL) and streptomycin (100 µg/mL), additionally containing 0.1% DMSO and the respective test compound or 0.1% DMSO alone as untreated control. After overnight (14–16 h) incubation, luminescence was measured using the Dual-Glo® Luciferase Assay System (Promega, Madison, WI, USA) according to the manufacturer's protocol with a Tecan Spark®

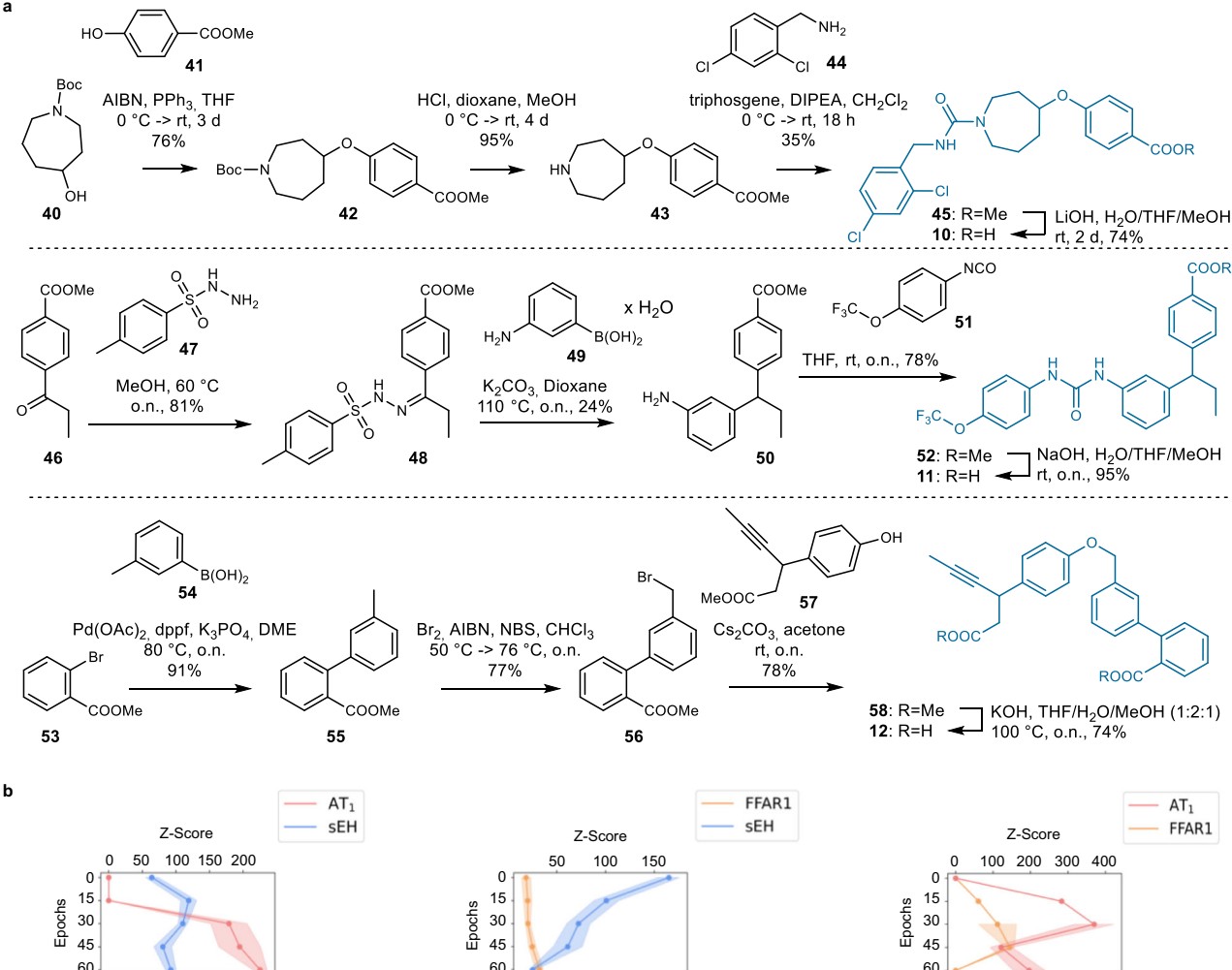

**Fig. 7 | Designed dual ligand candidates for AT₁/sEH (10), FFAR1/sEH (11) and FFAR1/AT₁ (12) from chemical language models. a** Synthesis of **10–12**. **b** Target prediction (Z-scores) of designs obtained by beam search over the fine-tuning procedure using the Similarity Ensemble Approach (SEA)[39] for the target of interest. Z-Scores are the mean ± standard deviation (SD). For each epoch beam search designs (width 50) were generated and only valid SMILES were analyzed. Source data are provided as a Source Data file.

10 M multimode microplate reader. Relative light units (RLU) for normalization of transfection efficiency and cell growth were calculated by dividing firefly luminescence by renilla luminescence and multiplication by 1000. Fold activation was obtained by dividing the mean RLU of a test compound at a respective concentration by the mean RLU of the untreated control. All samples were tested in duplicates in at least three biologically independent experiments. For dose response curve fitting and calculation of the $EC_{50}/IC_{50}$ values, the equation "[Agonist]/[Inhibitor] vs. response – Variable slope (four parameters)" was used in GraphPad Prism 7. The following pFA-CMV-hNR-LBD clones and reference ligands were used: THRβ (pFA-CMV-hTHRβ-LBD[71], 1 μM T3), PPARδ (pFA-CMV-hPPARδ-LBD[72], 1 μM L-165,041), FXR (pFA-CMV-hFXR-LBD[73], 1 μM GW4064).

## Chemistry
**General.** Unless stated otherwise, reactions were performed in dry, pre-heated Schlenk glassware under argon atmosphere. Chemicals purchased from BLD and Merck were of reagent grade and used without further purification. Dry solvents were purchased from Sigma-Aldrich and stored under argon atmosphere. Other solvents−especially for work-up procedures−were of reagent grade or purified by distillation (*iso*-hexanes, EtOAc). Reactions were monitored by thin layer chromatography (TLC) on TLC Silica gel 60 F₂₅₄ aluminum sheets

by Merck and visualized under fluorescent light (254 or 366 nm) or by ceric ammonium molybdate staining (5% $(NH_4)_6Mo_7O_{24} \times 4\ H_2O$, 0.2% $Ce(SO_4)_2$ in 5%iger $H_2SO_4$). Reaction and intermediate product control was performed by mass spectrometry on an Advion Interchim MS system with APCI using either ASAP for solutions or an Advion Plate express for TLC. Manual normal phase flash column chromatography was performed under positive $N_2$ pressure using silica 60 M from Macherey-Nagel. Automated normal phase flash column chromatography (CC) was performed on an Interchim Puriflash XS520Plus using corresponding PF-15/30/50SIHP columns. NMR spectroscopy was performed on a Bruker BioSpin AV400 or AV500. Chemical shifts ($\delta$) are reported in ppm relative to tetramethylsilane (TMS) or the residual solvent signal protons. Signal multiplicity is reported as singlet (s), broad singlet (bs), doublet (d), triplet (t), quartet (q), quintet (quin), heptet (hept), multiplet (m), doublet of doublets (dd), triplet of triplets (tt), doublet of triplets (dt), triplet of doublets (td). Coupling constants ($J$) are given in Hz. Assignments were made by means of two-dimensional experiments ($^1H$-$^1H$-COSY, $^1H$-$^{13}C$-HMQC, $^1H$-$^{13}C$-HMBC). Quantitative $^1H$ NMR (qHNMR) were acquired according to a method described by Pauli et al.[74] with internal calibration. The qHNMR measurements were carried out under conditions allowing complete relaxation to assure the exact determination of peak area ratios. High-resolution mass (HRMS) spectra were recorded on an MALDI LTQ

**Table 3 | Activity of dual ligand designs 10–12 on the targets of interest**

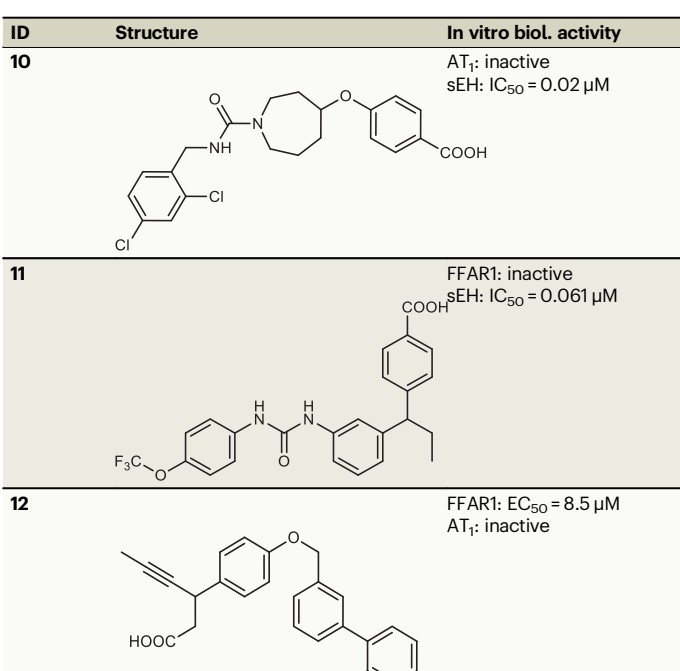

| ID | Structure | In vitro biol. activity |
|---|---|---|
| 10 | | $AT_1$: inactive<br>sEH: $IC_{50} = 0.02\ \mu M$ |
| 11 | | FFAR1: inactive<br>sEH: $IC_{50} = 0.061\ \mu M$ |
| 12 | | FFAR1: $EC_{50} = 8.5\ \mu M$<br>$AT_1$: inactive |

The dual $AT_1$/sEH (**10**), FFAR1/sEH (**11**) and FFAR1/$AT_1$ (**12**) ligand designs modulated only one of the targets of interest. Data are the mean; $n \geq 3$.

ORBITRAP XL instrument (Thermo Fisher Scientific Inc., Waltham, USA). using a α-cyano-4-hydroxycinnamic acid (HCCA) matrix or on a Thermo Finnigan LTQ FT Ultra FT-ICR (ESI). LCMS analysis was performed on a LCMS 2020 from Shimadzu (Duisburg, Germany). For analytical determination, a Luna 10u C18(2) (250 × 4.6 nm) and for semi-preparative purification, a Luna 10µ C18(2) (250 × 21.20 nm) column from Phenomenex LTD Deutschland (Aschaffenburg, Germany) was used. The system is equipped with a SPD 20 A UV/VIS detector ($\lambda = 240/280$ nm) and an ESI-TOF (measuring in the positive- and/or negative-ion mode). Mixtures of MeCN/0.1% aqueous formic acid were used as mobile phase with a flow rate of 0.1 mL/min (Scout Column) or 21 mL/min (semi-preparative) at rt. The following methods were used: Method 1: linear gradient from 50 to 90% MeCN over 10 min, 90% MeCN for 5 min, linear gradient from 90 to 50% MeCN over 1 min, 50% for 2 min; Method 2: 5% MeCN over 2 min, linear gradient from 5% to 90% MeCN over 12 min, 90% MeCN for 6 min, linear gradient from 90% to 5% MeCN over 1 min, 50% for 4 min; Purity of the compounds was determined by integrating the peaks of the UV-chromatogram. All compounds for biological testing had >95% purity according to qHNMR or LCMS analysis.

Synthetic procedures and analytical characterization of the test compounds are described in the Supporting Information.

### Reporting summary
Further information on research design is available in the Nature Portfolio Reporting Summary linked to this article.

## Data availability
All data generated in this study and supporting the results of this study are provided in the Supplementary Information and Source Data file. Source data are provided with this paper.

## Code availability
Code used in this study is available at Zenodo (https://doi.org/10.5281/zenodo.12795470)[75].

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

## Acknowledgements

The authors thank Lilia Weizel for technical support and Dr. Kerstin Hiesinger for sharing HK91. This research was funded by the European Union (ERC, NeuRoPROBE, 101040355). Views and opinions expressed are however those of the author(s) only and do not necessarily reflect those of the European Union or the European Research Council. Neither the European Union nor the granting authority can be held responsible for them. This work has received funding from the Innovative Medicines Initiative 2 Joint Undertaking (JU) under grant agreement No. 875510. The JU receives support from the European Union's Horizon 2020 research and innovation program, EFPIA, Ontario Institute for Cancer Research, Royal Institution for the Advancement of Learning McGill University, Kungliga Tekniska Hoegskolan, and Diamond Light Source Limited. J.H.M.E. and E.P. thank the Deutsche Forschungsgemeinschaft (DFG, SFB1039, TP A07) for financial support.

## Author contributions

L.I. and T.H. performed the computations; E.S., K.S., F.F.L., B.H., J.B., and J.P. synthesized the computational designs; E.S., J.H.M.E., and J.A.M. characterized the computational designs; L.I., T.H., J.A.M., J.P., E.P., and D.M. analyzed and interpreted the data; D.M. conceived and supervised the study; L.I. and D.M. prepared the figures and wrote the manuscript with contributions from all authors.

## Funding

## Competing interests

The authors declare no competing interests.
