## [Peer Review File · Nature Communications]

Automated design of multi-target ligands by generative deep learningREVIEWER COMMENTS

Reviewer #1 (Remarks to the Author):

In the present manuscript the authors describe the application the application of chemical language models to design dual-target ligands, and test some of the model predictions, showing several successful designs.

Overall, the methodology is sound, and well described so that it could also be applied by the readership. Overall I would recommend that the paper should eventually be published after a few points and questions (see below) have been addressed.

Questions/Suggestions

1) Were the most similar known ligands found in chembl in the original pre-training set of the CLM?

2) Did any of the (most similar) known ligands exhibit dual activity already, or was it just not reported before for these compounds? It also seems that the authors rediscover/re-test molecules from earlier publications (according to chembl), this should be made explicit e.g. *J. Med. Chem.* 2017, 60, 18, 7703–7724 - I don't think this is a problem, but similarity to known molecules has in the past been used as an argument against generative models (e.g. by <https://www.nature.com/articles/s41587-020-0418-2>). Whether this is justified depends on opinion, yet a lot of medicinal chemistry depends on careful, but small changes to the structures to explore the SAR and eventually achieve success, and having generative AI models do that is impressive.

3) The separation between table 1 and 2 in the supporting information should be made clearer (please create a new page for table 2). I would also suggest to describe in more detail what the header represents, in particular: Abundance (%), and Similarity FXR / sEH. Also the authors should indicate more clearly that molecules with entry in the ID field are the eventually selected ones.

4) The key work which introduced the CLM method is not cited
<https://arxiv.org/abs/1701.01329> - this should be additionally cited.

Minor Details:

- It is nice to have pictures of molecules within the experimental procedures in the SI
- It's also desirable to deposit pictures of the NMR spectra, or even the raw data in the SI

Reviewer #2 (Remarks to the Author):

The study objective of Merk and colleagues aiming to design dual-target ligands using CLMs is meaningful and the study concept exploring pairwise combinations of six targets from different families implicated in the multi-factorial metabolic syndrome is well thought out.

For three combinations of these targets with available chemically similar active compounds, dual-target compounds were confirmed after synthesis of three prioritized CLM candidates each; for three other target combinations with chemically dissimilar ligands, a single candidate was synthesized, but only active against one of the targets of each pair. The new single-/dual-target compounds had significantly varying potency.

A few general points for consideration:

- (a) It is difficult to draw any relevant conclusions concerning the “validity” of a de novo design approach on the basis of only 1-3 compounds that were synthesized and tested. While additional synthesis is not requested, the apparent success rate might only be rationalized if compound selection strictly depended on prior knowledge (see below), which would likely be controversially viewed.
- (b) The fine-tuning sets in Table S1 only contain a few compounds for each target; in one case (AT1) apparently only a single (!) reference. If so, this would be rather unusual and might call the relevance of fine-tuning into question (see below). Rationalizing this unusual choice (and possible consequences) would be highly recommended.
- (c) The strategy of combining fine-tuning sets proposed to consist of single-target compounds for each individual combination is certainly reasonable. However, at least for

the three target combinations with similar active compounds (and apparent design success), it cannot be ruled out that fine-tuning compounds might already have desired dual-target activity (but have not been tested against both targets), which should be taken into consideration.

While the work is technically sound the significance of the results, as presented, is difficult to judge, both from a medicinal chemistry and CLM design perspective (which is central to the study), for the following reasons:

- (i) Candidate selection from the 5000 sampled CLM compounds is heavily influenced/biased by prior knowledge of compound binding modes (exploited via docking) and pharmacophore elements (as reported in the manuscript).
- (ii) CLM dual-target compound design is successful when chemical space of actives closely overlaps for given target combinations and fails when the known compounds are dissimilar. Hence, it remains unclear if simple similarity methods would not produce results comparable to (or better than) CLMs.

Thus, a few controls will be essential to further examine the potential novelty of the findings and putative methodological advance of the de novo design approach.

- (1) Similarity: For each newly identified active compound, include side-by-side comparisons with the most similar fine-tuning compound in the SI.
- (2) Similarity: For the three target combinations for which dual-target compounds were identified, search ChEMBL/BindingDB for known dual-target compounds, report their numbers, and show comparisons of new and most similar known dual-target compounds.
- (3) Knowledge-biased selection: For each target combinations, rank the CLM output compounds only by sampling frequency and CLM likelihood estimates and report the ranks of the selected candidate compounds.
- (4) Fine-tuning sensitivity: For the three target combinations for which dual-target compounds were identified, pool all fine tuning sets, sample 5000 compounds, and screen the samples using (ligand- or target-based) pharmacophore models of each target for dual-target hits. Determine all candidates passing pharmacophore pairs, compare them with the original target pair-based compound samples, and identify the five nearest neighbors using

similarity calculations (predictivity of the pharmacophore models should be demonstrated and positive result, if obtained, be reported in the SI).

With programs such as MOE the authors are using, building such pharmacophore models is straightforward and requires only little efforts. As a form of “scrambling” control, (4) is expected to provide some insights whether or not the calculations are for the most similarity-driven or if the CLM indeed learns target-specific SAR features (from only very few structures) and successfully merges them in the generation of dual-target compounds. Further controls could be easily added, if needed.

Reviewer #3 (Remarks to the Author):

The author utilized a language model to train on molecular generation using the ChEMBL dataset and then fine-tuned it on a small dataset targeting binding activity, successfully biasing the model towards designing drug-like molecules similar to known ligands of the targeted pairs. The author also conducted dimensionality reduction comparisons on active small molecule data for different targets, to observe whether the active molecules for each target resemble those of another target. A higher similarity indicates that the active molecules of these two targets reside within a sub-space. Subsequently, the author combined the high-similarity pairs FXR/sEH, FXR/THR β , and PPAR δ /sEH into three sets of dual-target datasets to train the generation of active small molecules targeting both sites. For each dual-target combination, the author generated 5000 molecules, based on the frequency of generation and similarity to active molecules, initially screened out 12 molecules. These 12 molecules were then docked with proteins. Ultimately, 3 candidate molecules were screened out. The authors then synthesized these molecules and conducted biological experiments to demonstrate the effectiveness of their method. In terms of the small molecule generation algorithm, the author employed an existing method; however, their strategy for analyzing and merging target molecular data is something I have not seen in similar works. The entire work is very complete, with both computational and experimental validations being quite thorough.

However, there are still some aspects of the paper that are worthy of discussion.

1: The author used an LSTM model for molecule generation. Although LSTM is indeed an excellent model, in the realm of sequence generation, newer models like BERT,

Transformer, and GPT have already been proven powerful in many areas. Why did the author not consider these more recent models?

2: The paper begins with a dimensionality reduction comparison of molecular data from different targets before selecting similar datasets to merge. Subsequently, the paper carries out the generation of molecules trained for two targets, which is an interesting strategy. However, this seems to reduce the algorithm's generality, and considering AI generative models have shown remarkable results in areas like imaging, could this restriction in the paper also limit potential unexpected discoveries?

3: In the paper, 5000 molecules are generated for dual-targets, and 12 molecules are initially screened based on the frequency of generation and similarity to known active molecules. These are then subjected to molecular docking. The evaluation process used mainly assesses the similarity to known active molecules, with a lesser focus on docking results. Could the lesser effectiveness on the AT1/sEH, FFAR1/sEH, and AT1/FFAR1 target sets be related to this selection and evaluation process?

4: In figure 2f, the authors state that with the increase in finetuning epochs, the generated molecules increasingly resemble the molecular datasets for the two targets. However, it is not clearly visible in the graph that molecules generated at the same epoch appear in both target regions; instead, they appear only in a single area. This makes the figure somewhat confusing.

5: In figure 2g, the authors use a method called SEA to obtain Z-SCORES to demonstrate that, as finetuning epochs increase, the generated molecules increasingly resemble the molecular datasets for the two targets. From the graph, the final two curves do not converge; one rises while the other falls. Does this suggest that the finetuning method used by the authors ultimately still tends to bias toward one of the targets?

6: The authors have evaluated many properties of the generated results, but there is a lack of evaluation on the uniqueness and novelty, which is also very important for a generative model.

Reviewer #4 (Remarks to the Author):

Isigkeit and colleagues present a study focusing on the design of dual target-directed ligands using deep learning. This research holds significant relevance in the realm of multitarget

molecular design, given the historical challenge associated with conceiving and optimizing druglike Multitarget Directed Ligands (MTDLs). Demonstrating the utility of deep learning-based generative tools in addressing this specific challenge and producing hit compounds with multitarget activity while maintaining physicochemical properties akin to those of single-target molecules represents a noteworthy accomplishment with potential implications for the field.

The manuscript is well-crafted and effectively communicates its main points.

1)

In addition to t-SNE plots, it would be valuable to provide a more detailed analysis of any existing overlap in chemical space between molecules annotated for activity at single targets, if applicable.

2)

Traditionally, multitarget ligands are discussed in terms of linked, merged, or fused structures. While linked compounds may not be pertinent here, could the generated dual target derivatives be more accurately described as merged or fused? Furthermore, complementing the analysis based on Tanimoto similarity between Morgan fingerprints with methods such as RascalMCS (Raymond et al., *The Computer Journal*, 2002, 45(6), as implemented in RDKit) would enhance the comprehensiveness of the study.

3)

Have the authors explored the possibility of applying the generative approach to design FXR/PPARdelta dual-acting compounds, and subsequently comparing the proposed molecules to actual compounds previously reported by some of the authors (e.g., *J. Med. Chem.* 2020, 63, 15, 8369–8379)?

REVIEWER COMMENTS

Reviewer #1 (Remarks to the Author):

In the present manuscript the authors describe the application of chemical language models to design dual-target ligands, and test some of the model predictions, showing several successful designs.

Overall, the methodology is sound, and well described so that it could also be applied by the readership. Overall I would recommend that the paper should eventually be published after a few points and questions (see below) have been addressed.

We thank the Reviewer very much for evaluating our manuscript, for the positive feedback and for the constructive input. The Reviewer's comments were very valuable to improve the study and its presentation. We have addressed all the Reviewer's suggestions as outlined below.

Questions/Suggestions

1) Were the most similar known ligands found in chembl in the original pre-training set of the CLM?

Revised. We thank the Reviewer for this remark. We have clarified this point and added further comparison of the designed dual ligands with the most similar ligands in ChEMBL (Supplementary Table 6) and in the fine-tuning sets (Figure 5a, Supplementary Figure 3).

2) Did any of the (most similar) known ligands exhibit dual activity already, or was it just not reported before for these compounds? It also seems that the authors rediscover/re-test molecules from earlier publications (according to chembl), this should be made explicit e.g. J. Med. Chem. 2017, 60, 18, 7703–7724 - I don't think this is a problem, but similarity to known molecules has in the past been used as an argument against generative models (e.g. by <https://www.nature.com/articles/s41587-020-0418-2>). Whether this is justified depends on opinion, yet a lot of medicinal chemistry depends on careful, but small changes to the structures to explore the SAR and eventually achieve success, and having generative AI models do that is impressive.

Revised. We thank the Reviewer very much for this important remark. The fine-tuning compounds were carefully chosen to be selective for their bioactivity over the second target of the respective design approach according to literature and public bioactivity repositories (ChEMBL, etc.). We fully agree, however, that this may also mean that they were just not tested. To provide further evidence that the dual activity was indeed due to pharmacophore fusion by the CLM and not "encoded" in the templates, we tested the activity of commercially available fine-tuning molecules for PPAR δ /sEH as representative target pair. All evaluated fine-tuning molecules were confirmed selective for their annotated bioactivity over the respective second target (cf. Figure 5b) indicating that the dual activity was designed by the CLM. Nevertheless, a possible dual activity cannot be experimentally excluded for all fine-tuning templates, and we have added this consideration to the discussion. We thank the Reviewer very much for suggesting this important validation.

3) The separation between table 1 and 2 in the supporting information should be made clearer (please create a new page for table 2). I would also suggest to describe in more detail what the header represents, in particular: Abundance (%), and Similarity FXR / sEH. Also the authors should indicate more clearly that molecules with entry in the ID field are the eventually selected ones.

Revised. We thank the Reviewer for this comment. The presentation of all Supplementary Tables has been improved. Tables have been separated more clearly and column headers are better explained in the updated version.

4) The key work which introduced the CLM method is not cited <https://arxiv.org/abs/1701.01329> - this should be additionally cited.

Revised. We thank the Reviewer for this remark. The missing reference has been added.

Minor Details:

- It is nice to have pictures of molecules within the experimental procedures in the SI

Revised. We have added the chemical structures to the synthetic procedures in the SI as suggested.

- It's also desirable to deposit pictures of the NMR spectra, or even the raw data in the SI

Revised. The NMR spectra of all final compounds have been added to the SI as suggested.

Reviewer #2 (Remarks to the Author):

The study objective of Merk and colleagues aiming to design dual-target ligands using CLMs is meaningful and the study concept exploring pairwise combinations of six targets from different families implicated in the multi-factorial metabolic syndrome is well thought out.

For three combinations of these targets with available chemically similar active compounds, dual-target compounds were confirmed after synthesis of three prioritized CLM candidates each; for three other target combinations with chemically dissimilar ligands, a single candidate was synthesized, but only active against one of the targets of each pair. The new single-/dual-target compounds had significantly varying potency.

We thank the Reviewer very much for evaluating our manuscript and for the critical but very helpful and constructive feedback which has significantly contributed to improving our study and the technical validation of our results. We have addressed all the Reviewer's comments as outlined below.

A few general points for consideration:

(a) It is difficult to draw any relevant conclusions concerning the “validity” of a de novo design approach on the basis of only 1-3 compounds that were synthesized and tested. While additional synthesis is not requested, the apparent success rate might only be rationalized if compound selection strictly depended on prior knowledge (see below), which would likely be controversially viewed.

Revised. We thank the Reviewer for this important comment. We fully agree that the study is "underpowered" to reveal a success rate of the multi-target design approach using CLM but this was also not our intention. We aimed to probe and validate the prospective application of CLM for designed polypharmacology and evaluate whether this approach can yield designed dual ligands (without focusing on a success rate which will highly depend on the target combination of interest). We have rephrased the manuscript to clarify these points and avoid any misunderstandings.

(b) The fine-tuning sets in Table S1 only contain a few compounds for each target; in one case (AT1) apparently only a single (!) reference. If so, this would be rather unusual and might call the relevance of fine-tuning into question (see below). Rationalizing this unusual choice (and possible consequences) would be highly recommended.

Revised. We thank the Reviewer for this comment. The fine-tuning sets were balanced over all targets and always contained at least five template molecules per target (also for AT₁) but the presentation of the AT₁ fine tuning set was confusing due to a page break. We have resolved this issue by restructuring Supplementary Table 1.

(c) The strategy of combining fine-tuning sets proposed to consist of single-target compounds for each individual combination is certainly reasonable. However, at least for the three target combinations with similar active compounds (and apparent design success), it cannot be ruled out that fine-tuning compounds might already have desired dual-target activity (but have not been tested against both targets), which should be taken into consideration.

Revised. We thank the Reviewer for this important remark. The fine-tuning molecules were carefully chosen based on their annotated bioactivity to avoid inclusion of any template that already possesses the intended dual activity but we fully agree with the Reviewer that the respective compounds may just not have been tested. Thus, for further validation, we have tested the commercially available fine-tuning molecules for PPAR δ /SEH as representative target pair on both targets in the same assays used for characterization of the dual ligands (Figure 5b). We found all tested fine-tuning molecules to be selective over the second target of the respective design approach providing further evidence that the dual activity was not already "encoded" in the templates but designed by the CLM. Nevertheless, not all fine-tuning

molecules can be tested, and a potential bias of "unknown" dual ligands cannot be fully excluded. This consideration has been added to the discussion. We thank the Reviewer very much for requesting this important validation.

While the work is technically sound the significance of the results, as presented, is difficult to judge, both from a medicinal chemistry and CLM design perspective (which is central to the study), for the following reasons:

(i) Candidate selection from the 5000 sampled CLM compounds is heavily influenced/biased by prior knowledge of compound binding modes (exploited via docking) and pharmacophore elements (as reported in the manuscript).

(ii) CLM dual-target compound design is successful when chemical space of actives closely overlaps for given target combinations and fails when the known compounds are dissimilar. Hence, it remains unclear if simple similarity methods would not produce results comparable to (or better than) CLMs.

We thank the Reviewer very much for these critical remarks and the valuable suggestions for further validation (below). This extended analysis has clearly improved the study. We have added the requested comparisons and performed the suggested additional experiments as discussed below.

Thus, a few controls will be essential to further examine the potential novelty of the findings and putative methodological advance of the de novo design approach.

(1) Similarity: For each newly identified active compound, include side-by-side comparisons with the most similar fine-tuning compound in the SI.

Revised. Side-by-side comparison of the designed dual ligands 1-9 with the most similar fine-tuning compounds has been added in Figure 5a and Supplementary Figure 3. We compared traditional Tanimoto similarity computed on Morgan fingerprints and RascalMCES.

(2) Similarity: For the three target combinations for which dual-target compounds were identified, search ChEMBL/BindingDB for known dual-target compounds, report their numbers, and show comparisons of new and most similar known dual-target compounds.

Revised. The numbers of dual ligands of the target combinations annotated in ChEMBL have been added to the manuscript (Figure 2a). Despite using a very loose activity cutoff of 100 μ M, the numbers of known dual ligands are very low (FXR/sEH = 34; FXR/THR β = 1; PPAR δ /sEH = 0).

(3) Knowledge-biased selection: For each target combinations, rank the CLM output compounds only by sampling frequency and CLM likelihood estimates and report the ranks of the selected candidate compounds.

Revised. We thank the Reviewer very much for this suggestion. The sampling frequency and corresponding ranks have been added to Supplementary Table 4. Reporting CLM likelihood would not be meaningful here as we have used a series of five consecutive epochs for sampling and likelihood may vary dramatically between epochs while the sampling frequency can be reported as sum over the evaluated epochs.

(4) Fine-tuning sensitivity: For the three target combinations for which dual-target compounds were identified, pool all fine tuning sets, sample 5000 compounds, and screen the samples using (ligand- or target-based) pharmacophore models of each target for dual-target hits. Determine all candidates passing pharmacophore pairs, compare them with the original target pair-based compound samples, and identify the five nearest neighbors using similarity calculations (predictivity of the pharmacophore models should be demonstrated and positive result, if obtained, be reported in the SI).

With programs such as MOE the authors are using, building such pharmacophore models is straightforward and requires only little efforts. As a form of “scrambling” control, (4) is expected to provide some insights whether or not the calculations are for the most similarity-driven or if the CLM indeed learns target-specific SAR features (from only very few structures) and successfully merges them in the generation of dual-target compounds. Further controls could be easily added, if needed.

Revised. We thank the Reviewer very much for this valuable suggestion. The proposed analysis has been added (cf. Figure 5c-e) and further supports the pooled fine-tuning approach for dual ligand design by CLM. As suggested, we have trained the CLM with the fine-tuning molecules for four targets (FXR, THR β , PPAR δ and sEH) and used pharmacophore models to estimate the success rates of dual ligand design. Both the native baseline CLM (without fine-tuning) and the CLM trained on four targets designed substantially less molecules matching the pharmacophores for the target pairs of interest than the focused dual-target CLM. This was further supported by target prediction which revealed very low scores for the targets of interest for designs obtained from the baseline CLM and no consistent and balanced improvement by pooled fine-tuning for all four targets.

Reviewer #3 (Remarks to the Author):

The author utilized a language model to train on molecular generation using the ChEMBL dataset and then fine-tuned it on a small dataset targeting binding activity, successfully biasing the model towards designing drug-like molecules similar to known ligands of the targeted pairs. The author also conducted dimensionality reduction comparisons on active small molecule data for different targets, to observe whether the active molecules for each target resemble those of another target. A higher similarity indicates that the active molecules of these two targets reside within a sub-space. Subsequently, the author combined the high-similarity pairs FXR/sEH, FXR/THR β , and PPAR δ /sEH into three sets of dual-target datasets to train the generation of active small molecules targeting both sites. For each dual-target combination, the author generated 5000 molecules, based on the frequency of generation and similarity to active molecules, initially screened out 12 molecules. These 12 molecules were then docked with proteins. Ultimately, 3 candidate molecules were screened out. The authors then synthesized these molecules and conducted biological experiments to demonstrate the effectiveness of their method. In terms of the small molecule generation algorithm, the author employed an existing method; however, their strategy for analyzing and merging target molecular data is something I have not seen in similar works. The entire work is very complete, with both computational and experimental validations being quite thorough.

We thank the Reviewer very much for performing peer-review of our manuscript, for the positive feedback and for the valuable input. The Reviewer's suggestions have led to important improvements. We have addressed all the Reviewer's comments. Point-by-point answers are given below.

However, there are still some aspects of the paper that are worthy of discussion.

1: The author used an LSTM model for molecule generation. Although LSTM is indeed an excellent model, in the realm of sequence generation, newer models like BERT, Transformer, and GPT have already been proven powerful in many areas. Why did the author not consider these more recent models?
Revised. We thank the Reviewer for this important comment. CLM based on LSTM have shown strong performance in transfer learning from sets of template ligands for a target of interest even in low data scenarios which is a critical feature for application in de novo design of bioactive molecules where data on active templates is often limited. The core of this study was to extend the application of CLM to the challenging task of multi-target design. Achieving successful de novo design of dual ligands by machine learning would be very valuable to overcome the elaborate development of such compounds by systematic and rational approaches. We fully agree with the Reviewer that newer models hold great potential for de novo design but think that they should be evaluated in prospective applications on easier tasks first. We have added these considerations to the discussion section.

2: The paper begins with a dimensionality reduction comparison of molecular data from different targets before selecting similar datasets to merge. Subsequently, the paper carries out the generation of molecules trained for two targets, which is an interesting strategy. However, this seems to reduce the algorithm's generality, and considering AI generative models have shown remarkable results in areas like imaging, could this restriction in the paper also limit potential unexpected discoveries?

Revised. We thank the Reviewer very much for these interesting thoughts. It should be noted that the analysis by dimensionality reduction with tSNE was only used to evaluate the chemical space of ligands for the targets pairs of interest but not for the development of the dual target CLMs. As discussed in the manuscript, it offered potential insights for the high performance of the CLM on three target pairs (FXR/sEH, FXR/THR β , PPAR δ /sEH) while other combinations (AT $_1$ /sEH, FFAR1/sEH, AT $_1$ /FFAR1) were not successful. Nevertheless, we fully agree with the Reviewer that CLM and machine learning in general hold more potential to achieve the intended multi-target de novo design possibly requiring other

approaches than employed here. It should still be kept in mind that simultaneous modulation of many target combinations will not be achievable for too diverse binding sites preventing the binding of a common ligand. We have added these considerations to the discussion.

3: In the paper, 5000 molecules are generated for dual-targets, and 12 molecules are initially screened based on the frequency of generation and similarity to known active molecules. These are then subjected to molecular docking. The evaluation process used mainly assesses the similarity to known active molecules, with a lesser focus on docking results. Could the lesser effectiveness on the AT1/sEH, FFAR1/sEH, and AT1/FFAR1 target sets be related to this selection and evaluation process?

Revised. We thank the Reviewer for this important comment. The selection process of molecules designed by machine learning remains a challenge. Several studies have relied on external methods using docking- or similarity-based ranking and recent work has provided model-intrinsic approaches to design prioritization from CLM (e.g., perplexity, beam search, sampling frequency). Here we aimed to give strong weight to model intrinsic measures of design quality (beam search, sampling frequency) to (i) avoid external bias and (ii) obtain a method that is broadly applicable. Docking was only used for a small number of prioritized compounds after the actual selection process as last filter before elaborate synthesis and in vitro testing was performed. Analysis and comparison of dual ligand designs and fine-tuning templates (Figure 2c,d,i) indicated that the differently overlapping chemical spaces of (known) ligands could explain the varying success in obtaining dual ligands for the different target pairs. Nevertheless, the Reviewer is perfectly right that design selection is a crucial step in the de novo design process and may have affected the success rate. We have added these considerations to the discussion.

4: In figure 2f, the authors state that with the increase in finetuning epochs, the generated molecules increasingly resemble the molecular datasets for the two targets. However, it is not clearly visible in the graph that molecules generated at the same epoch appear in both target regions; instead, they appear only in a single area. This makes the figure somewhat confusing.

Revised. We thank the Reviewer for this remark. We have improved the presentation of Figure 2f (now 2i) to better illustrate that molecules designed by the CLM after dual target fine-tuning populate the interface of the chemical space regions of selective ligands for the two targets of interest in contrast to molecules designed by the baseline model (epoch 0).

5: In figure 2g, the authors use a method called SEA to obtain Z-SCORES to demonstrate that, as finetuning epochs increase, the generated molecules increasingly resemble the molecular datasets for the two targets. From the graph, the final two curves do not converge; one rises while the other falls. Does this suggest that the finetuning method used by the authors ultimately still tends to bias toward one of the targets?

We thank the Reviewer for this remark. We agree that variance of the score between epochs likely indicates that the bias of the CLM towards the respective target fluctuated as it can be also observed in the similarity to the fine-tuning sets (Figure 2f). However, compared to the baseline model (epoch 0), the Z-scores improved for all targets during fine-tuning indicating that the model captured relevant ligand features for all targets. To account for a certain variance between epochs, we used five consecutive epochs to sample de novo designs.

6: The authors have evaluated many properties of the generated results, but there is a lack of evaluation on the uniqueness and novelty, which is also very important for a generative model.

Revised. We thank the Reviewer for this important remark. The missing analysis of uniqueness and novelty has been added (cf. Supplementary Tables 2,3).

Reviewer #4 (Remarks to the Author):

Isigkeit and colleagues present a study focusing on the design of dual target-directed ligands using deep learning. This research holds significant relevance in the realm of multitarget molecular design, given the historical challenge associated with conceiving and optimizing druglike Multitarget Directed Ligands (MTDLs). Demonstrating the utility of deep learning-based generative tools in addressing this specific challenge and producing hit compounds with multitarget activity while maintaining physicochemical properties akin to those of single-target molecules represents a noteworthy accomplishment with potential implications for the field.

The manuscript is well-crafted and effectively communicates its main points.

We thank the Reviewer very much for evaluating our manuscript, for the constructive suggestions for improvement and for the positive feedback. The Reviewer's input was very valuable for improving the study and its presentation. We have addressed all comments as discussed below.

1)

In addition to t-SNE plots, it would be valuable to provide a more detailed analysis of any existing overlap in chemical space between molecules annotated for activity at single targets, if applicable.

Revised. We thank the Reviewer for this comment. We have extended the analysis of the known ligands of the targets of interest and their overlap and similarity. We have added pie charts showing the numbers of known ligands for the target pairs (Figure 2a) including the few known dual ligands (with loose activity cutoff $<100 \mu\text{M}$) and violin plots illustrating the pairwise Tanimoto similarity distribution between the known ligands for the target pairs (Figure 2b). In addition to the fingerprint-based tSNE, we have included tSNE plots based on CATS as fuzzy pharmacophore descriptors which revealed similar distribution and overlap (Figure 2c) and we have included a scaffold analysis (Figure 2d).

2)

Traditionally, multitarget ligands are discussed in terms of linked, merged, or fused structures. While linked compounds may not be pertinent here, could the generated dual target derivatives be more accurately described as merged or fused? Furthermore, complementing the analysis based on Tanimoto similarity between Morgan fingerprints with methods such as RascalMCS (Raymond et al., *The Computer Journal*, 2002, 45(6), as implemented in RDKit) would enhance the comprehensiveness of the study.

Revised. We thank the Reviewer very much for these valuable suggestions. Structural comparison of the designed dual ligands and the most similar selective ligands (Figure 4d) indicated that the pharmacophores were merged by the CLM. We have added this consideration to the manuscript. We have additionally analyzed the active dual ligands using RascalMCES (Figure 5a, Supplementary Figure 3) which provided another valuable illustration how structural features were combined by the CLM in the dual ligands.

3)

Have the authors explored the possibility of applying the generative approach to design FXR/PPAR δ dual-acting compounds, and subsequently comparing the proposed molecules to actual compounds previously reported by some of the authors (e.g., *J. Med. Chem.* 2020, 63, 15, 8369–8379)?

Revised. We thank the Reviewer very much for this valuable suggestion. We have extended the application of our multi-target design approach to the target combination FXR/PPAR δ as suggested and analyzed the performance based on the previously developed dual ligands. Using the same fine-tuning sets of selective FXR and PPAR δ ligands indeed resulted in the design of previously reported dual agonists and several structurally similar molecules. The results are presented in Supplementary Table 5.

REVIEWER COMMENTS

Reviewer #1 (Remarks to the Author):

I want to thank the authors for the additional work they put in the manuscript, and reasonable responses to the reviewers' comments.

In my opinion the manuscript is now ready for publication.

Reviewer #2 (Remarks to the Author):

The authors have thoroughly addressed the partly overlapping comments of the reviewers and have carried out suggested controls that help to put the design approach and results into perspective. Publication of the revised manuscript in its current form is recommended.

Reviewer #3 (Remarks to the Author):

After reading the revised paper and the response letter, I believe that the authors have addressed the review comments fairly well. However, there are still two minor issues:

1. should the positions of figures fig2j and fig2k be swapped?
2. Our question 3 actually raises the following concern: After the LSTM model is trained, it generates sequences based on their similarity. If the subsequent selection of generated sequences also mainly relies on sequence similarity, the criteria seem somewhat repetitive and monotonous.

Reviewer #4 (Remarks to the Author):

The authors have performed a commendable job in addressing all my previous comments in this revised version of the submitted manuscript.

REVIEWER COMMENTS

Reviewer #1 (Remarks to the Author):

I want to thank the authors for the additional work they put in the manuscript, and reasonable responses to the reviewers' comments.

In my opinion the manuscript is now ready for publication.

We thank the Reviewer for the very constructive feedback on our manuscript in the first round of peer-review and for supporting publication of the revised version.

Reviewer #2 (Remarks to the Author):

The authors have thoroughly addressed the partly overlapping comments of the reviewers and have carried out suggested controls that help to put the design approach and results into perspective. Publication of the revised manuscript in its current form is recommended.

We thank the Reviewer very much for the valuable input in the first round of peer-review and for recommending publication of the revised manuscript.

Reviewer #3 (Remarks to the Author):

After reading the revised paper and the response letter, I believe that the authors have addressed the review comments fairly well. However, there are still two minor issues:

We thank the Reviewer very much for re-evaluating our manuscript and for the valuable additional input. We have addressed the Reviewer's comments as outlined below.

1. should the positions of figures fig2j and fig2k be swapped?

Revised. We thank the Reviewer very much for spotting this error. The panels were indeed in the wrong order. Figure 2 has been corrected accordingly.

2. Our question 3 actually raises the following concern: After the LSTM model is trained, it generates sequences based on their similarity. If the subsequent selection of generated sequences also mainly relies on sequence similarity, the criteria seem somewhat repetitive and monotonous.

We thank the Reviewer for this valuable comment. Automated scoring of de novo designs from a chemical language model with model-intrinsic measures (without external techniques) is desirable as it reduces the external bias. Based on the ability of chemical language models to learn the conditional probability for any character in a SMILES string considering preceding characters, beam search (Moret et al. Angew. Chem. Int. Ed. Engl. 2021, 60, 19477-19482), perplexity (Moret et al., J. Chem. Inf. Model. 2022, 62, 1199-1206) and sampling frequency (Ballarotto et al., J. Med. Chem. 2023, 66, 8170-8177) have been identified as suitable internal design prioritization approaches to overcome the need for external scoring. Therefore, we consider the uniformity of using sequences for training and model-intrinsic sequence-based measures for design prioritization as a strength. We have added a sentence in the manuscript to clarify this point.

Reviewer #4 (Remarks to the Author):

The authors have performed a commendable job in addressing all my previous comments in this revised version of the submitted manuscript.

We thank the Reviewer very much for the constructive comments on the original manuscript and for the positive feedback on the revised version.

REVIEWERS' COMMENTS

Reviewer #3 (Remarks to the Author):

Thank you to the author and editor, I have no further questions.